# Explaining Concept Shift with Interpretable Feature Attribution

Ruiqi Lyu [1]   Alistair Turcan [1]   Bryan Wilder [1]

## Abstract

Concept shift occurs when the distribution of labels conditioned on the features changes between domains, which can make even a well-tuned ML model miscalibrated on a new domain. Identifying these shifted features provides unique insight into how feature-label relationships differ between domains, considering the difference may be across a scientifically relevant dimension, such as time, disease status, population, etc. In this paper, we propose SGShift, a method for attributing performance degradation under concept shift in tabular data to a sparse set of shifted features. We frame concept shift as a feature selection task to learn the features that can explain performance differences between models in the source and target domain. This framework enables SGShift to adapt powerful statistical tools such as generalized additive models, knockoffs, and absorption towards identifying these shifted features. We conduct extensive experiments in synthetic and real data across various ML models and find SGShift can identify shifted features much more accurately than baseline methods, requires few samples in the shifted domain, and is robust to complex cases of concept shift.

## 1. Introduction

Machine learning (ML) models are often trained on vast amounts of data, but will inevitably encounter test distributions that differ from the training set. Such distribution shift is one of the most common failure modes for ML in practice. When models do fail, model developers need to diagnose and correct the problem. In the simplest case, this may simply consist of gathering more data to retrain the model. However, in other cases, it may be necessary to fix issues in an underlying data pipeline, add new features to replace

ones that have become uninformative, or undertake other more complex interventions. A necessary starting point for any such process is to understand what changed in the new dataset. Developing such understanding may even have scientific importance. For instance, a novel virus variant may emerge with new risk factors, lowering the performance of models that predict disease progression, or specific mutations in the genome could have differing relevance to disease between ancestries, weakening polygenic risk score models due to fundamentally different biology between populations (Duncan et al., 2019; Martin et al., 2019).

We propose methods for diagnosing distribution shift, focusing specifically on the case of *concept shift*, or when the conditional distribution of the label given the features, $p(Y \mid X)$, differs between the source and target distribution. Concept shift represents the difficult case where the relationship between features and outcome has changed, as opposed to marginal covariate or label shifts impacting only $p(X)$ or $p(Y)$ by themselves. Indeed, (Liu et al., 2024) document concept shift as the primary contributor to performance degradation across a wide range of empirical examples of distribution shifts, and (Gama et al., 2014) describe it as affecting "most of the real world applications". In this setting, our goal is to understand the features that led $p(Y \mid X)$ to differ between the source and target domains.

Understanding distribution shift has been the subject of increasing interest. However, existing methods mostly operate relative to a structure for the data which is prespecified by the analyst, for example a known causal graph (Zhang et al., 2023; Subbaswamy et al., 2021), fixed decomposition of the variables (Singh et al., 2024), or particular assumed models for distribution shift in which out-of-distribution performance can be identified using only unlabeled data (Chen et al., 2022). Methods that do not impose such structural conditions largely repurpose other tools to explain distribution shift as a secondary objective. For example Mougan et al. (2023) propose to look for changes in model explanations, while Liu et al. (2023) fit a decision tree to explain differences in predictions from source and target domain models as part of a larger empirical investigation.

We introduce SGShift, a new method directly designed for diagnosing concept shift. SGShift offers robust statistical performance, particularly with limited target-domain sam-

[1]Carnegie Mellon University. Correspondence to: Ruiqi Lyu <ruiqil@andrew.cmu.edu>.

*Proceedings of the 43rd International Conference on Machine Learning*, Seoul, South Korea. PMLR 306, 2026. Copyright 2026 by the author(s).

ples and without requiring prespecified causal structure. Just as sparsity is an effective principle for learning predictive models in many settings due to sparse mechanism shift (Schölkopf et al., 2021), we hypothesize that the *update* needed to adapt a source-domain model to the target domain may often be well approximated by a sparse function of the features (a fact that we empirically verify in several application domains). In this case, a useful explanation of concept shift is to identify a small set of features that drive the change between the two distributions, which could e.g. be the subject of potential modeling fixes. SGShift frames this problem as learning an update to a source distribution's predictive model using a minimal set of features to recover the performance loss in the target distribution. We show how this formulation allows simple, principled, and easily implemented diagnoses of distribution shift, without requiring any prior knowledge, causal information, or parametric priors regarding the dataset.

We benchmark SGShift against several baselines on semi-synthetic datasets with known feature shifts, observing greatly superior performance at identifying concept shifted features (referred to as shifted features throughout). We then apply SGShift to two real-data settings and recover real-world concept shifts consistent with findings from medical and biological literature, such as respiratory failure's reduced association with COVID-19 hospitalization after Omicron and ancestry-associated rare variants related to Lupus prediction. Together, these findings provide evidence that SGShift can recover accurate and interpretable descriptions of concept shift across a wide range of settings.

### 1.1. Additional Related Work

**Covariate shift.** Much of the existing work on distribution shift has focused on the setting where the marginal feature distribution $P(X)$ changes while the conditional distribution $P(Y \mid X)$ remains unchanged, i.e., covariate shift. For instance, (Kulinski et al., 2020) introduce statistical tests to identify which variables have shifted between source and target domains, while (Kulinski & Inouye, 2023) propose explaining observed shifts via a learned transportation map between the source and target distributions, not distinguishing between features and labels. $P(X)$ shift can be identified by methods like two-sample tests (Jang et al., 2022) or classifiers (Lipton et al., 2018) and corrected by techniques such as importance sampling (Sugiyama et al., 2007). Cai et al. (2023) further use these ideas to correct covariate shift by regarding the unexplained residual as a shift in $P(Y \mid X)$, although they don't correct or explain the concept shift. Although these methods can be effective for addressing covariate shift, they often do not delve into potential shifts in the conditional distribution. Explaining shifts in $P(Y \mid X)$ typically involves performing feature-by-feature analyses of the conditional distribution $P(Y \mid X_i)$ (Guidotti

et al., 2018). However, such univariate assessments risk detecting spurious shifts due to unadjusted confounding in the presence of collinearity among predictors (Raskutti et al., 2010). Kulinski & Inouye (2023) consider an unsupervised setting where the goal is to identify a set of features whose distribution differs (e.g., sensors that have been compromised by an adversary), as opposed to identifying features whose relationship with a supervised label has changed.

**Conditional distribution shift.** Recent efforts have begun to tackle shifts in the conditional distribution $P(Y \mid X)$ more directly. For example, (Zhang et al., 2023) consider changes in a causal parent set as a whole, relying on known causal structures. (Mougan et al., 2023) propose a model-agnostic "explanation shift detector" that applies SHAP (Shapley additive explanations) to a source-trained model and covariates in both source and target domains, without including the outcomes in the target domain. They then use a two-sample test on the feature-attribution distributions from SHAP to detect whether the model's decision logic has changed because of the changing of $P(X)$ across domains. Despite its effectiveness in signaling shifts, this approach does not pinpoint which features are driving the changes in $P(Y \mid X)$. (Singh et al., 2024) decompose the domain loss gap into predefined marginal and conditional segments, then allocate feature-level contributions, while (Singh et al., 2025) automatically discover subgroups within the data for which to produce feature-level explanations. (Subbaswamy et al., 2021) stress tests a source model before distribution shift, requiring a prespecified set of shifting variables. (Chen et al., 2022) focus on estimation of performance shift on an unlabeled dataset, but this require restrictive assumptions for identifiability, particularly that non-shifted features have no shifts at all when conditioned on the shifted features and label between datasets. WhyShift (Liu et al., 2023) compares two independently trained models - one from each domain - and analyze their difference to locate regions of covariate space with the largest predictive discrepancy. SGShift differs in that we aim to explicitly identify what the features contributing to conditional distribution shift are without requiring any prior knowledge of the dataset.

## 2. Preliminaries and Problem Formulation

**Problem formulation.** We consider the problem of identifying a set of features that can represent an observed conditional distribution shift. We observe labeled i.i.d. samples $\{(X_i^{(S)}, Y_i^{(S)})\}_{i=1}^{n_S} \sim P_S$ and $\{(X_i^{(T)}, Y_i^{(T)})\}_{i=1}^{n_T} \sim P_T$ from source and target domains, respectively, where $X_i \in \mathbb{R}^p$ and $n_S$ and $n_T$ are the number of samples in the source and target domain. A source model $h_S : \mathbb{R}^p \to \mathbb{R}$ is trained on the source domain and evaluated in the target domain. Concept shift occurs when $P_T(Y \mid X = x) \neq P_S(Y \mid X = x)$ for some covariate values $x$. Let

$\mu_S(x) = \mathbb{E}_S[Y \mid X = x], \mu_T(x) = \mathbb{E}_T[Y \mid X = x]$. On a chosen modeling scale, for example, the link scale for a generalized linear model, define the conditional-response contrast $\Delta_g(x) = g(\mu_T(x)) - g(\mu_S(x))$. Our primary goal is to identify a small set of raw feature indices $A \subseteq [p]$ such that $\Delta_g(x)$ is well approximated by a function depending only on $x_A$.

The difficulty is that the source and target samples are unpaired: we only observe labels from $P_S(Y \mid X)$ for source-domain covariates and labels from $P_T(Y \mid X)$ for target-domain covariates. Accordingly, it is not possible to directly apply existing methods for sparse regression. The most directly related work, the WhyShift framework introduced by (Liu et al., 2023) for diagnosing concept shift, takes a plugin approach. A plugin strategy first fits models on the two datasets separately to approximate $\mathbb{E}_S[Y \mid X]$ and $\mathbb{E}_T[Y \mid X]$. Second, it fits a second model regressing some difference metric of $\widehat{\mathbb{E}}_S[Y \mid X]$ and $\widehat{\mathbb{E}}_T[Y \mid X]$ on $X$ to summarize the structure in $\Delta$. However, this plugin approach risks an accumulation of errors, particularly when we are interested in recovering structure related to sparsity: given noisy approximations to the two conditional expectations, the difference between $\widehat{\mathbb{E}}_S[Y \mid X]$ and $\widehat{\mathbb{E}}_T[Y \mid X]$ will not necessarily display the same sparsity pattern as $\Delta$ (as we observe experimentally). It is also potentially challenging when we have limited target-domain data, since separately fitting $\mathbb{E}_T[Y \mid X]$ may be especially difficult in this setting.

## 3. Method

Our method, SGShift, circumvents these difficulties by reformulating the problem so that existing sparse regression and controlled-selection methods can be applied to a correction term. Instead of fitting separate models for $\mathbb{E}_S[Y \mid X]$ and $\mathbb{E}_T[Y \mid X]$ and then estimating their difference, SGShift starts with a source-domain model $h_S(X)$ and learns a sparse target-domain *correction*

$$\min_{\hat{\Delta}} \mathbb{E}_T[\ell(h_S(X) + \hat{\Delta}(X), Y)] \quad \text{s.t. } \hat{\Delta} \text{ is } k\text{-sparse.}$$

This formulation treats $h_S(X)$ as a fixed offset and learns only the correction from target-domain data. Our theory gives estimation consistency for the sparse correction under standard high-dimensional M-estimation assumptions, while SGShift-K provides false-discovery control for selected features through knockoffs. Exact support recovery would require stronger beta-min and design assumptions, and is not claimed for the basic SGShift estimator.

### 3.1. SGShift: Instantiation with $\ell_1$ regularization

Our suggested implementation of SGShift uses a generalized additive correction with $\ell_1$ regularization. Let $h_S(X)$

denote the fixed source-domain predictor on the same modeling scale as the link function $g$. We model

$$g(\mathbb{E}_T[Y \mid X]) = h_S(X) + \phi(X)^\top \delta \qquad (1)$$

Here, $g$ is a link function, $\phi : \mathbb{R}^p \to \mathbb{R}^K$ is a fixed basis map. Raw-feature attributions are then obtained by grouping basis coordinates according to the raw features on which they depend. The default choice is the standardized identity basis, so $K = p$ and $\phi(X) = X$ after standardization. When richer but still interpretable corrections are desired, $\phi$ can include low-order terms such as pairwise interactions. Appendix K suggests that SGShift is reasonably robust to both under- and over-specified interaction bases. In order to control the sparsity level of $\delta$, SGShift solves the averaged target-domain penalized loss

$$\hat{\delta} = \arg \min_{\delta \in \mathbb{R}^K} \left\{ L(\delta) + \lambda \|\delta\|_1 \right\}$$

where $L(\delta) := \frac{1}{n_T} \sum_{i=1}^{n_T} \ell\left( h_S(X_i^{(T)}) + \phi(X_i^{(T)})^\top \delta, Y_i^{(T)} \right).$

(2)

where $\ell(\eta, y)$ is the per-sample negative log-likelihood evaluated at linear predictor $\eta$. By default, we choose $\lambda$ by cross-validation on target-domain loss when the goal is predictive correction. When shifted-feature selection is the main goal, we vary $\lambda$ to obtain a ranking across sparsity levels or use the heuristic in Appendix F. Empirically, performance was stable when varying $\lambda$ from 0.1 to 10 times the selected value.

### 3.2. SGShift-A: Refined fitting considering source model misspecification

Prioritizing shifted features relies on an existing model trained on the source dataset. However, it may be that this model does not represent the data well due to difficulties in model fitting. To reduce the extent to which source-model misfit biases the selection of shifted features, we incorporate an additional absorption term. The main absorption idea is a modeling approximation: we assume that a component of source-model misspecification is sufficiently shared across source and target domains that it can be represented by a common correction term, while the remaining target-specific correction is attributed to conditional shift.

Let $\phi_i^{(S)} = \phi(X_i^{(S)}) \in \mathbb{R}^K$ and $\phi_i^{(T)} = \phi(X_i^{(T)}) \in \mathbb{R}^K$. SGShift-A solves:

$$(\hat{\omega}, \hat{\delta}) = \arg \min_{\omega, \delta \in \mathbb{R}^K} \left\{ L^A(\omega, \delta) + \lambda_\omega \|\omega\|_1 + \lambda_\delta \|\delta\|_1 \right\}, \quad (3)$$

where

$$L^A(\omega, \delta) = w_S \frac{1}{n_S} \sum_{i=1}^{n_S} \ell\left( h_S(X_i^{(S)}) + (\phi_i^{(S)})^\top \omega, \ Y_i^{(S)} \right)$$

$$+ w_T \frac{1}{n_T} \sum_{i=1}^{n_T} \ell\left( h_S(X_i^{(T)}) + (\phi_i^{(T)})^\top(\omega + \delta), \right.$$

$$\left. Y_i^{(T)} \right).$$

(4)

Here, $\omega$ is a shared correction and $\delta$ is target-specific, $w_S = w_T = 1/2$ by default so that source and target losses contribute comparably even when the sample sizes differ. We induce hierarchical regularization $\lambda_\omega < \lambda_\delta$ so that variation explainable by a shared correction is absorbed before introducing target-specific shifted features. In practice, we select $\lambda_\delta$ using the same strategy as SGShift, either cross-validation on target-domain loss or the heuristic in Appendix F. We set $\lambda_\omega = c\lambda_\delta$ with $c < 1$; in our experiments, values of $c$ between $0.1$ and $0.9$ gave similar behavior, and we use this range as a default sensitivity grid.

### 3.3. SGShift-K: Explicit false discovery control with knockoffs

While $\ell_1$ regularization enables sparse correction learning, its selected support can include false discoveries, especially when features are correlated or the target sample is small. For shifted-feature selection, we adapt the Model-X knockoffs framework (Candes et al., 2018). Knockoffs generate synthetic features that preserve the dependence structure of the original features while being conditionally unrelated to the response, enabling controlled variable selection. Let $\tilde{X} = [\tilde{X}^{(1)}, \ldots, \tilde{X}^{(p)}] \in \mathbb{R}^{n_T \times p}$ denote a knockoff copy of $X_T$ and let $[\phi_T \ \tilde{\phi}_T] = [\phi(X_T) \ \phi(\tilde{X})] \in \mathbb{R}^{n_T \times 2K}$. For the formal knockoff guarantee, the tested units should be interpreted as raw features or as pre-specified groups of basis functions associated with raw features. If $\phi$ includes interaction or nonlinear basis terms, then either a valid knockoff construction for the transformed design must be used, or the knockoff statistics should be grouped at the raw-feature level. We fit the correction model using both original and knockoff basis functions:

$$\hat{\delta}' = \arg \min_{\delta' \in \mathbb{R}^{2K}} \left\{ L^K(\delta') + \lambda \|\delta'\|_1 \right\}, \quad (5)$$

where

$$L^K(\delta') = \frac{1}{n_T} \sum_{i=1}^{n_T} \ell\Big( h_S(X_i^{(T)})$$

$$+ [\phi(X_i^{(T)}) \ \phi(\tilde{X}_i^{(T)})]\delta', Y_i^{(T)} \Big). \quad (6)$$

Write $\delta' = (\delta, \tilde{\delta})$, where $\delta$ corresponds to original basis functions and $\tilde{\delta}$ to knockoff basis functions. We compute

knockoff importance statistics $W_j$ comparing each original feature to its knockoff copy. To reduce randomization variability across knockoff draws, we then apply a repeated-knockoff aggregation procedure for feature selection. For stability selection, we threshold empirical selection frequencies across knockoff draws; for nominal FDR control of the aggregated set, we use the e-value derandomized knockoff construction and e-BH procedure (Ren & Barber, 2024). Notably, the objective of SGShift-K is shifted feature selection only with the generation of knockoff copies, while SGShift and SGShift-A can do simultaneous feature selection and target model correcting from the trained source model.

Appendix D gives the construction details, and Appendix E states the assumptions under which the stability-thresholded and e-BH aggregated procedures control errors.

### 3.4. Theoretical guarantees

We show that when the model in Equation 1 is well-specified, SGShift has desirable theoretical guarantees for estimating of the sparse correction coefficients $\delta$ under proper choice of the regularization parameter $\lambda$. Importantly, this only requires imposing assumptions on the form of the between-distribution difference $\Delta_g$, rather than on the complete regression function $\mathbb{E}_T[Y \mid X]$, which is allowed to be nonparametric (as opposed to the standard Lasso setting), while consistency still requires standard high-dimensional assumptions on the target design, loss curvature, and sparse correction class. In particular, we obtain the following:

**Theorem 3.1** (Estimation guarantee for SGShift). *Suppose $g\left(\mathbb{E}_T[Y \mid X]\right) = h_S(X) + \phi(X)^\top \delta^*$ for some $\delta^* \in \mathbb{R}^K$ with basis-coordinate support $A \subseteq [K]$ and $|A| = a$. Let $L$ be the averaged loss in Equation 2. Assume the loss is convex and differentiable in $\delta$, the rows $\phi(X_i^{(T)})$ are sub-Gaussian, and $L$ satisfies local cone-restricted strong convexity (RSC, justification in Appendix A) around $\delta^*$ over the cone $\mathcal{C}(A) = \{u \in \mathbb{R}^K : \|u_{A^c}\|_1 \leq 3\|u_A\|_1\}$. If $\lambda \geq 2\|\nabla L(\delta^*)\|_\infty$ and $\lambda \asymp \sqrt{\frac{\log K}{n_T}}$, then, with high probability, $\|\hat{\delta} - \delta^*\|_2^2 \lesssim a\lambda^2 \lesssim \frac{a \log K}{n_T}$.*

The proof is in Appendix B. Here, $\lesssim$ means asymptotically bounded above up to a constant factor, and $\asymp$ means asymptotically the same order up to constant factors.

An analogous result holds for SGShift-A under the corresponding augmented-design assumptions.

**Theorem 3.2** (Estimation guarantee for SGShift-A). *Let $\beta^* = (\omega^*, \delta^*) \in \mathbb{R}^{2K}$ be the population minimizer of the SGShift-A risk, with support sizes $s_\omega$ and $s_\delta$. Define augmented rows $z_i^{(S)} = (\phi(X_i^{(S)}), 0)$, $z_i^{(T)} = (\phi(X_i^{(T)}), \phi(X_i^{(T)}))$. Assume the augmented empirical loss is convex and satisfies local cone-restricted strong convexity with constant $\kappa_A > 0$ over the weighted $\ell_1$*

*cone induced by the supports of $\omega^*$ and $\delta^*$. If $\lambda_\omega \geq 2\|\nabla_\omega L^A(\beta^*)\|_\infty$, $\lambda_\delta \geq 2\|\nabla_\delta L^A(\beta^*)\|_\infty$, then, with high probability, $\|\hat{\omega} - \omega^*\|_2^2 + \|\hat{\delta} - \delta^*\|_2^2 \lesssim \frac{s_\omega \lambda_\omega^2 + s_\delta \lambda_\delta^2}{\kappa_A^2}$. When $w_S, w_T$ are fixed constants and the augmented design satisfies the corresponding restricted-eigenvalue condition, one may take $\lambda_\omega, \lambda_\delta \asymp \sqrt{\log K / \min(n_S, n_T)}$ up to constants, with sharper coordinate-specific rates possible when the two blocks are analyzed separately.*

Further, the use of knockoffs in SGShift-K allows us to obtain error-control guarantees for false selections.

**Theorem 3.3** (Stability Selection Control for SGShift-K). *Let $A^c = \{k : \delta_k^* = 0\}$ denote the set of correction-null tested coordinates or feature groups with zero coefficient in the true data distribution and $B$ the number of knockoff samples.*

*(PFER Control) Assume for each $k \in A^c$, the events $\{k \in \hat{A}^{[b]}\}_{b=1}^B$ are independent across repeats and satisfy $P(k \in \hat{A}^{[b]}) \leq \alpha$ uniformly over $b$, where $\alpha$ is an assumed per-repeat false-selection probability bound, $\hat{A}^{[b]}$ is the selected set from the bth knockoff repeat, and $\hat{A}(\pi)$ is the stability-thresholded set across all repeats. For any stability threshold $\pi > \alpha$:*

$$\mathbb{E}\left[\left|\hat{A}(\pi) \cap A^c\right|\right] \leq |A^c| \exp\left(-2B(\pi - \alpha)^2\right).$$

*(FDR Control) If the repeated knockoff runs are aggregated using the derandomized-knockoff e-value construction and the e-BH procedure of Ren & Barber (2024), then the final selected set controls FDR at the target level $q$ under the Model-X knockoff assumptions and the correction-null sign-flip property for the statistics $W_k^{[b]}$.*

Theorem 3.2 guarantees per family error rate (PFER) control for the stability-thresholded set and false discovery rate (FDR) control for the e-BH aggregated knockoff set under their respective assumptions. The proof is in Appendix E. We also provide a heuristic one-pass penalty-calibration rule for SGShift in Appendix F.

## 4. Experiments

**Evaluation setup** We evaluate our method on three real-world healthcare datasets (details in Appendix G) split in such a way to exhibit natural concept shifts, 30-day Diabetes Readmission (Strack et al., 2014) split by ER admission, COVID-19 Hospitalizations (of Us Research Program Investigators, 2019) split by pre and post-Omicron, and SUPPORT2 Hospital Expenses (Connors et al., 1995) split by death in hospital. For each of these 3 naturally shifted datasets, we construct semi-synthetic simulations, consistent with previous work (Singh et al., 2025; Zhang et al., 2023). We fit a "generator" model" to the real labels in

source domain, relabeling the source data, then simulate the target dataset's labels with an induced conditional shift by perturbing $g(E[Y \mid X])$ based on selected input features. A "base" model is then trained from the relabeled source domain. We vary base and generator models to be each combination of decision tree, logistic/linear regression, gradient boosting, and support-vector machines, for a total of 16 settings in each dataset and 48 total settings. We consider 3 scenarios in each setting, **sparse shift**, where a small set of features are shifted, **dense shift**, where $> 60\%$ of the features are shifted, and **sample size**, the sparse setting but with varying the amount of samples in the target domain. All features to shift are selected randomly. We construct additional synthetic simulations that allow us to isolate the impact of more complex shift cases by simulating 1000 samples in each domain alongside a specific distribution of features with the same base/generator setup as in the semi-synthetic case. This includes **high dimensional shifts**, where 100-500 features exist and 20% are shifted, **feature correlation**, varying the maximum feature correlation $\rho$ from 0.1 to 0.9, with i-th and j-th predictors correlated as $\rho^{|i-j|}$ with 500 features and 20% are shifted, and **signal-to-noise**, varying the signal-to-noise ratio (SNR) from 1 to 16 with maximum feature correlation 0.7, 200 features, and 20% are shifted. In all simulations, a feature ranking is obtained by varying the penalty parameter from 0.0001 to 100, and feature selection performance (a binary 0/1 label) measured on this ranking with AUC and recall at false positive rate 5%.

**Baselines** We consider 3 baseline models which also use both features and labels in source and target domain to identify shifted features. **Diff**, a method we construct where we simply compute the outcome discrepancies of two "base models" separately trained on source and target data, and apply sparse regression on held-out samples and the base models' outcome probability differences to identify features contributing to the shifts. **WhyShift** (Liu et al., 2023) uses two "base models" separately trained on source and target domains and computes model outcome probability discrepancies, then trains a non-linear decision tree on these discrepancies to detect regions (paths in the tree) responsible for conditional shifts. We extract the features from any path in the learned tree with feature importance $> 0$ and consider them as the shifted features. **SHAP**, a Shapley value-based method we adapt from (Mougan et al., 2023) such that we can find individual features that differ between datasets. SHAP trains "base models" separately on source and target data, computes the Shapley value of each feature, and ranks the largest absolute differences between models.

### 4.1. Benchmarking

**Accuracy in detecting shifted features.** First, we examine the case of sparse shifts, in line with SGShift's sparsity assumption (Table 1). Across model settings and datasets,

| Model Match | Diff | WhyShift | SHAP | SGShift | SGShift-A | SGShift-K |
|---|---|---|---|---|---|---|
| **Sparse simulations** | | | | | | |
| **Diabetes Readmission** | | | | | | |
| Matched | $0.64 \pm 0.09$ | $0.73 \pm 0.08$ | $0.77 \pm 0.12$ | $0.80 \pm 0.01$ | $0.81 \pm 0.01$ | **$0.90 \pm 0.01$** |
| Mismatched | $0.69 \pm 0.06$ | $0.72 \pm 0.04$ | $0.76 \pm 0.04$ | $0.79 \pm 0.03$ | $0.81 \pm 0.04$ | **$0.86 \pm 0.04$** |
| **COVID-19** | | | | | | |
| Matched | $0.78 \pm 0.05$ | $0.76 \pm 0.06$ | $0.81 \pm 0.10$ | $0.86 \pm 0.02$ | $0.88 \pm 0.03$ | **$0.99 \pm 0.02$** |
| Mismatched | $0.77 \pm 0.03$ | $0.71 \pm 0.05$ | $0.77 \pm 0.03$ | $0.85 \pm 0.03$ | $0.80 \pm 0.03$ | **$0.97 \pm 0.03$** |
| **SUPPORT2** | | | | | | |
| Matched | $0.83 \pm 0.05$ | $0.67 \pm 0.06$ | $0.82 \pm 0.09$ | $0.92 \pm 0.01$ | $0.94 \pm 0.01$ | **$0.96 \pm 0.01$** |
| Mismatched | $0.80 \pm 0.03$ | $0.67 \pm 0.03$ | $0.76 \pm 0.05$ | $0.86 \pm 0.01$ | $0.88 \pm 0.01$ | **$0.95 \pm 0.01$** |
| **Dense simulations** | | | | | | |
| **Diabetes Readmission** | | | | | | |
| Matched | $0.54 \pm 0.09$ | $0.52 \pm 0.08$ | $0.64 \pm 0.12$ | $0.71 \pm 0.01$ | $0.72 \pm 0.02$ | **$0.86 \pm 0.01$** |
| Mismatched | $0.58 \pm 0.06$ | $0.57 \pm 0.04$ | $0.60 \pm 0.04$ | $0.72 \pm 0.04$ | $0.71 \pm 0.03$ | **$0.82 \pm 0.04$** |
| **COVID-19** | | | | | | |
| Matched | $0.79 \pm 0.05$ | $0.65 \pm 0.06$ | $0.86 \pm 0.10$ | $0.80 \pm 0.02$ | $0.83 \pm 0.02$ | **$0.95 \pm 0.02$** |
| Mismatched | $0.78 \pm 0.03$ | $0.74 \pm 0.05$ | $0.78 \pm 0.03$ | $0.76 \pm 0.02$ | $0.76 \pm 0.03$ | **$0.93 \pm 0.03$** |
| **SUPPORT2** | | | | | | |
| Matched | $0.62 \pm 0.05$ | $0.56 \pm 0.06$ | $0.62 \pm 0.09$ | $0.89 \pm 0.01$ | $0.89 \pm 0.01$ | **$0.92 \pm 0.01$** |
| Mismatched | $0.73 \pm 0.03$ | $0.60 \pm 0.03$ | $0.70 \pm 0.05$ | $0.88 \pm 0.01$ | $0.89 \pm 0.01$ | **$0.92 \pm 0.01$** |

*Table 1.* **Performance in identifying shifted features.** AUC for detecting the true shifted features in sparse (top) and dense (bottom) semi-synthetic simulations. Matched refers to when generator and base model are the same, mismatched when they differ. Results are aggregated across the 4 matched and 12 mismatched settings. 95% confidence intervals are evaluated across configurations.

SGShift-K achieves the strongest performance and each variant of SGShift strongly outperforms baselines Diff, WhyShift and SHAP, with AUC typically greater than 0.9, 0.1-0.2 higher than the nearest baseline. Despite the presence of model mismatch, SGShift variants still attain high performance in the mismatched setting, on average only 0.02 AUC below the matched setting.

We next examine the case of dense concept shift, violating SGShift's sparsity assumption. Table 1 shows evaluation results. Despite the assumption of sparsity, SGShift-K still attains AUC greater than 0.8 and 0.9, and all variants of SGShift still broadly outperform baselines. Baseline methods generally experienced a substantial performance drop in the dense case, such as all methods in the Diabetes dataset, each with AUC around 0.6, down from around 0.75 previously. SGShift variants are robust towards dense shift and do not over-emphasize a few features when many may be shifted. While perhaps counterintuitive given the sparsity assumption, SGShift likely performs well as it effectively acts as a regularized feature-ranking procedure, and can still capture most of the signal even when most shift coefficients are nonzero. Additional experiments testing global and interaction shifts are available in Appendix J and K, respectively.

Next, we vary the sample size available in the target domain, simulating an online learning setting where data is gradually streaming in. Results for SUPPORT2 are reported in Figure 1. SGShift-K is able to identify over half the shifting features

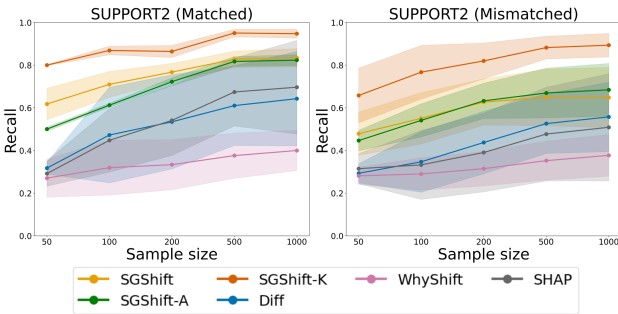

*Figure 1.* **Performance across sample sizes.** Sample size is varied from 50 to 1000. 95% CI's are shown across 16 simulation settings. Recall is measured at fixed FPR 5%.

given only 100 samples regardless the model setting, and over 85% given 500 samples. This indicates SGShift-K is indeed an effective diagnostic tool, not requiring many samples for accurate feature identification. Similar results are reported for Diabetes and COVID-19 in Appendix I.

We analyze more complex cases of concept shift in Figure 2. In the high dimensional case with 500 features and just 1000 samples in each domain, all variants of SGShift can still strongly detect shifted features with AUC> 0.9 in the matched case and > 0.8. This is in contrast with baseline methods, where performance collapses at the 500 feature mark. When features are exceedingly correlated, all variants of SGShift again achieve strong performance relative to baseline methods with average AUC> 0.8 in both cases.

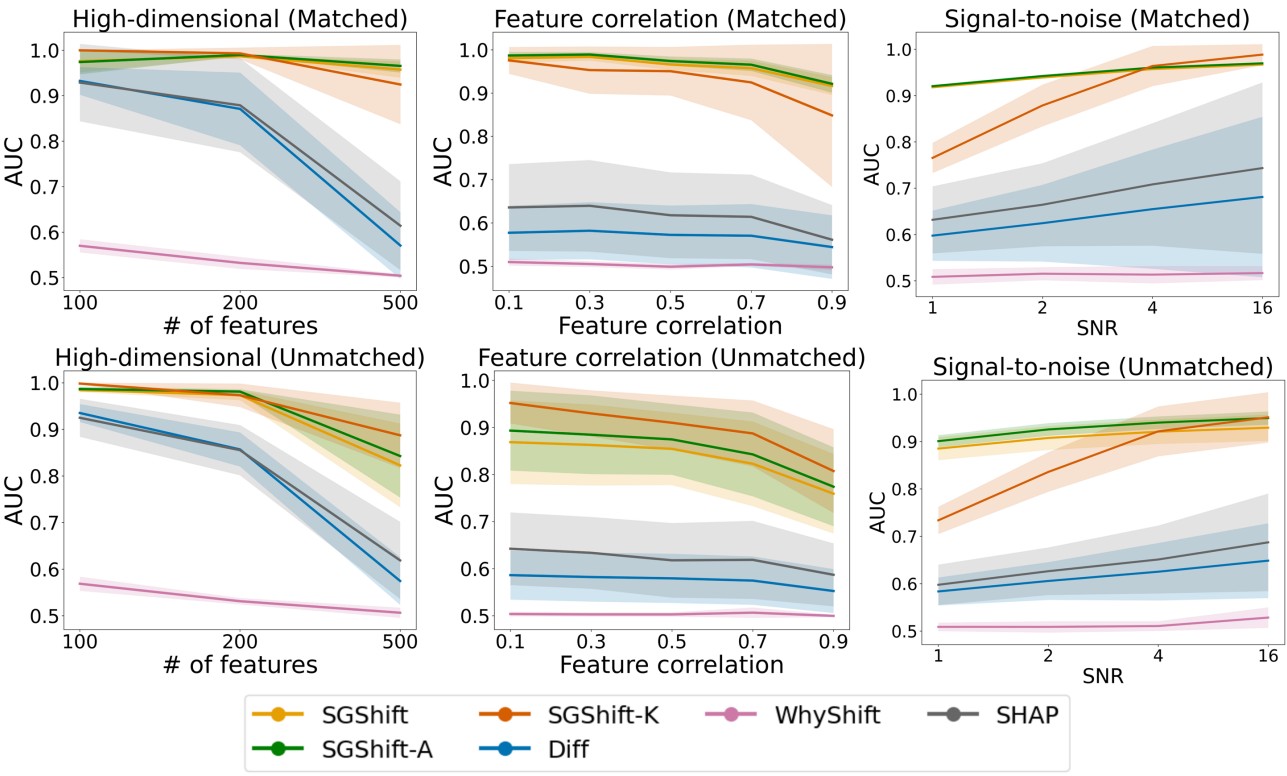

*Figure 2.* **Complex cases of concept shift.** Performance of each method at identifying shifted features in high dimensional (left), feature correlation (center), and SNR simulation settings. 95% CI's are shown across 16 simulation settings.

As the signal-to-noise ratio decreases substantially, SGShift and SGShift-A are relatively unaffected and maintain strong feature selection capability with AUC$>$ 0.9. SGShift-K loses power as the signal becomes exceedingly weak as the feature importance statistics calculated by knockoffs become noisier, but nevertheless still outperforms baselines by $>0.15$ AUC when SNR=1.

### 4.2. Real data

**Sparse predictive corrections in real data.** One advantage of SGShift is that it offers simultaneous performance recovery alongside feature attribution. In real data, this lets us test whether target-domain performance loss can be well approximated by a sparse update, rather than verifying that the underlying data-generating concept shift is itself sparse, where in contrast all baselines considered provide shifted feature estimates only. Operationally, we say that a target-domain performance drop admits a sparse update if 90% of the performance loss can be recovered using only a small set of features. Applied to the original data with natural concept shift, SGShift is able to learn updates to the source model that recover 90% of the performance loss in the target domain (Figure 3), requiring less than 1/3 of the total features, and in some cases as little as one feature. We sought to validate if this was the result of complex concept

shift or more simple covariate or label shift. We applied inverse propensity weighting (IPW) (Gardner et al., 2023), a common method for correcting for covariate shift to each of our model settings. We find that IPW can only recover at most 15% of the performance drop, and often has little change or even reduces model performance, likely due to insufficient sample size for the IPW procedure. We then apply IPW to the target labels to account for label shift, and find this explains only one case of shift and less than 25% of the shift on average. We conclude that covariate or label shift alone cannot explain these performance drops. As an illustrative example, we plot the performance recovery curve of SGShift for an SVM model in the SUPPORT2 dataset, varying the penalty parameter to allow more features to contribute to SGShift's source model update. With only 6 features, the performance loss can be completely recovered, whereas correcting for label or covariate shift offers little improvement. We next test for sparsity at scale, training models on cross-state splits of the ACS Income Datasets (Gardner et al., 2023) and identifying 87 cases of concept shift, where source-trained models underperform in the target domain. Applying SGShift to these cases, we find that all 87 cases can be recovered, many with less than 5% of features, and all required less than 50%. These results suggest that, in these real-data settings, target-domain per-

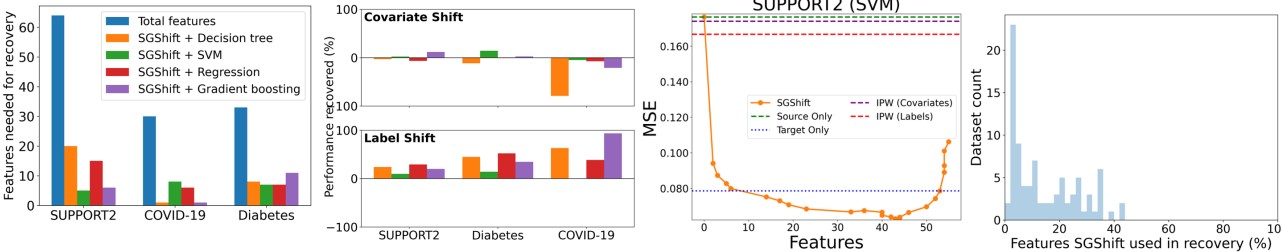

*Figure 3.* **Sparse predictive corrections in real world concept shifts.** A) Number of features SGShift required to learn a sparse correction to the source model that recovered 90% of the performance loss in the target domain. B) Performance recovery achieved by applying IPW to correct for covariate and label shift. C) By decreasing the feature penalization penalty to add more features to SGShift's update, we see how many terms are needed to recover performance in the target domain. D) Percent of features SGShift needs to correct for concept shift in 87 cases from the ACS income datasets.

formance degradation can often be well approximated by a sparse update.

**Case study in healthcare.** We next evaluate whether the top features selected by SGShift-K for the COVID-19 pre-/post-Omicron split are consistent with known clinical patterns (data split in Table 2). The highest ranked feature across all models is "respiratory failure" with a negative sign, consistent with the broad observation of reduced severe lower-respiratory disease during Omicron compared to the previous Delta variant (Adjei et al., 2022), partly due to Omicron's decreased ability at infecting lung cells (Hoffmann et al., 2023). More severe cases may be taking place in other pathways, such as the upper respiratory tract (Wickenhagen et al., 2025). Other selected features, including "abnormal breathing" and "other circulatory/respiratory signs," also have negative corrections, plausibly reflecting the same shift away from lower-respiratory severity. Features such as "primary diagnosis," "insurance claims," "chronic ischemic heart disease," and "age 45–64" may reflect that non-lung-related comorbidities contribute relatively more to hospitalization risk after the reduced role of severe lung involvement (Lewnard et al., 2022).

**Case study in genetics.** We consider a known case of concept shift in the difference in Lupus severity and prevalence between ancestries (Langefeld et al., 2017). We use the gene expression from 149 healthy and Lupus-affected Europeans, and 107 healthy and Lupus-affected Asians (Perez et al., 2022), and aim to predict Lupus status using the top 1000 variable genes in B cells, a cell type commonly implicated in Lupus. We split by ancestry and apply SGShift-K to find genes contributing to concept shift. Expectedly, we first observe an XGBoost model trained on Europeans underperforms when applied to Asians (European AUC 1.0, Asian AUC 0.84). SGShift-K discovers 6 genes in B cells contributing to this shift: ERRFI1, RP11-666A1.5, CTD-2561B21.11, AC012309.5, AC074212.5, and AP001059.5, all with negative coefficients. ERRFI1 and RP11-666A1.5 are both differentially expressed in B cells between these an-

cestries (Wang & Gazal, 2023). A genetic basis of difference in Lupus between ancestries has been discovered, and CTD-2561B21.11, AC012309.5, AC074212.5, and AP001059.5 are underpinned by eQTLs or repeat variants common in Europeans but rare in East Asians (Morris et al., 2016; Langefeld et al., 2017). Interferon signatures commonly correlate with Lupus prevalence, and Asians have elevated background interferon levels compared to Europeans, such as RP11-666A1.5 (Rector et al., 2023). These results suggest that SGShift-K selects biologically plausible features associated with ancestry-specific differences in Lupus prediction, although SGShift tests for predictive differences and is not designed to identify direct causal biological mechanisms.

## 5. Discussion

We have presented SGShift, a method for attributing concept shift between datasets to a sparse set of features. Our work contributes towards understanding what makes models fail between datasets. We prove statistical guarantees regarding SGShift's false discovery control and demonstrate high power in detecting true shifted features, even when the assumption of sparsity is violated. Our real-data results suggest that target-domain performance degradation in tabular prediction tasks can often be well approximated by sparse update. Future work could include optimizing model performance by explicitly modeling the difference between datasets given the identified shifted features, disentangling various contributors to concept shift such as label or measurement drift, or extending SGShift to non-tabular data, e.g., images or graphs.

## Acknowledgements

This material is based upon work supported by the United States of America Department of Health and Human Services, Centers for Disease Control and Prevention, under award numbers U01IP001121 and NU38FT000005; and

contract number 75D30123C1590. Any opinions, findings, and conclusions or recommendations expressed in this material are those of the author(s) and do not necessarily reflect the views of the United States of America Department of Health and Human Services, Centers for Disease Control and Prevention.

## Impact Statement

This paper presents work whose goal is to advance the field of Machine Learning. There are many potential societal consequences of our work, none which we feel must be specifically highlighted here.

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

# A. Justification of Restricted Strong Convexity (RSC)

The RSC condition is used only locally around the true correction parameter $\delta^*$ and only over the usual Lasso cone. We state it for the averaged loss in Equation 2.

**Lemma A.1** (Local cone-restricted RSC). *Let $\phi_i = \phi(X_i^{(T)}) \in \mathbb{R}^K$ be i.i.d. sub-Gaussian vectors whose population covariance satisfies a restricted-eigenvalue condition over the relevant sparse cones. Define*

$$L(\delta) = \frac{1}{n_T} \sum_{i=1}^{n_T} \ell\{h_S(X_i^{(T)}) + \phi_i^\top \delta, Y_i^{(T)}\}.$$

*Assume $\ell(\eta, y)$ is twice differentiable and has curvature bounded below by $\kappa > 0$ on a neighborhood of the linear predictors generated by $\delta^* + u$ for all $u$ in the local set $\mathcal{C}(A, r) = \{u : \|u_{A^c}\|_1 \leq 3\|u_A\|_1, \|u\|_2 \leq r\}$. Then, for sufficiently large $n_T$, with high probability, for all $u \in \mathcal{C}(A, r)$, we have $L(\delta^* + u) - L(\delta^*) - \langle u, \nabla L(\delta^*) \rangle \geq c\|u\|_2^2$, for a constant $c > 0$ depending on $\kappa$, the population restricted-eigenvalue constant, and the sub-Gaussian norm of $\phi_i$.*

*Proof.* By Taylor's theorem,

$$L(\delta^* + u) - L(\delta^*) - \langle u, \nabla L(\delta^*) \rangle = \frac{1}{2} u^\top \nabla^2 L(\tilde{\delta}) u$$

for some $\tilde{\delta}$ on the line segment between $\delta^*$ and $\delta^* + u$. The Hessian satisfies

$$\nabla^2 L(\tilde{\delta}) = \frac{1}{n_T} \sum_{i=1}^{n_T} \nabla_\eta^2 \ell\{h_S(X_i^{(T)}) + \phi_i^\top \tilde{\delta}, Y_i^{(T)}\} \phi_i \phi_i^\top \succeq \kappa \hat{\Sigma},$$

where $\hat{\Sigma} = n_T^{-1} \sum_i \phi_i \phi_i^\top$. Under the population restricted-eigenvalue condition, standard restricted-eigenvalue concentration for sub-Gaussian designs gives

$$u^\top \hat{\Sigma} u \geq c' \|u\|_2^2$$

uniformly over the local Lasso cone when $n_T \gtrsim a \log K$. Combining the curvature bound and the restricted-eigenvalue bound yields the result. $\square$

# B. Proof of Theorem 3.1: Estimation Guarantee for SGShift

*Proof.* Let $d = \hat{\delta} - \delta^*$ and $A = \mathrm{supp}(\delta^*)$. By optimality of $\hat{\delta}$ in Equation 2,

$$L(\hat{\delta}) + \lambda\|\hat{\delta}\|_1 \leq L(\delta^*) + \lambda\|\delta^*\|_1.$$

Rearranging and adding/subtracting the first-order term gives

$$L(\hat{\delta}) - L(\delta^*) - \langle d, \nabla L(\delta^*)\rangle \leq -\langle d, \nabla L(\delta^*)\rangle + \lambda(\|\delta^*\|_1 - \|\hat{\delta}\|_1).$$

If $\lambda \geq 2\|\nabla L(\delta^*)\|_\infty$, then

$$-\langle d, \nabla L(\delta^*)\rangle \leq \frac{\lambda}{2}\|d\|_1.$$

Using decomposability of the $\ell_1$ norm and the fact that $\delta^*_{A^c} = 0$,

$$\|\delta^*\|_1 - \|\hat{\delta}\|_1 \leq \|d_A\|_1 - \|d_{A^c}\|_1.$$

Therefore,

$$L(\hat{\delta}) - L(\delta^*) - \langle d, \nabla L(\delta^*)\rangle \leq \frac{3\lambda}{2}\|d_A\|_1 - \frac{\lambda}{2}\|d_{A^c}\|_1.$$

Since $L$ is convex, the Bregman divergence on the left-hand side is nonnegative. Therefore,

$$\|d_{A^c}\|_1 \leq 3\|d_A\|_1.$$

Applying Lemma A.1 and using $\|d_A\|_1 \leq \sqrt{a}\|d\|_2$, we obtain

$$c\|d\|_2^2 \leq \frac{3\lambda}{2}\|d_A\|_1 \leq \frac{3\lambda}{2}\sqrt{a}\|d\|_2.$$

Thus $\|d\|_2 \lesssim \sqrt{a}\lambda$, and hence $\|\hat{\delta} - \delta^*\|_2^2 \lesssim a\lambda^2$. For sub-Gaussian designs and exponential-family losses,

$$\|\nabla L(\delta^*)\|_\infty \lesssim \sqrt{\frac{\log K}{n_T}}$$

with high probability by standard concentration. Taking $\lambda \asymp \sqrt{\frac{\log K}{n_T}}$ gives

$$\|\hat{\delta} - \delta^*\|_2^2 \lesssim \frac{a\log K}{n_T}.$$

$\square$

**Remark.** The fixed offset $h_S$ does not affect the RSC argument because the optimization is over $\delta$; the curvature is determined by the loss and the basis design. The result establishes estimation consistency, not exact support recovery.

# C. Proof Sketch for Theorem 3.2: Estimation Guarantee for SGShift-A

Let $d_\omega = \hat{\omega} - \omega^*$ and $d_\delta = \hat{\delta} - \delta^*$, and let $S_\omega = \text{supp}(\omega^*)$, $S_\delta = \text{supp}(\delta^*)$. The optimality inequality for the weighted penalty gives

$$L^A(\hat{\omega}, \hat{\delta}) + \lambda_\omega \|\hat{\omega}\|_1 + \lambda_\delta \|\hat{\delta}\|_1 \leq L^A(\omega^*, \delta^*) + \lambda_\omega \|\omega^*\|_1 + \lambda_\delta \|\delta^*\|_1.$$

If $\lambda_\omega \geq 2\|\nabla_\omega L^A(\beta^*)\|_\infty$, $\lambda_\delta \geq 2\|\nabla_\delta L^A(\beta^*)\|_\infty$, the same decomposability argument yields a weighted cone condition. Applying local RSC for the augmented design gives

$$\|\hat{\omega} - \omega^*\|_2^2 + \|\hat{\delta} - \delta^*\|_2^2 \lesssim \frac{s_\omega \lambda_\omega^2 + s_\delta \lambda_\delta^2}{\kappa_A^2}.$$

# D. Construction and selection with Knockoffs

Following (Candes et al., 2018), a Model-X knockoff copy $\tilde{X} = [\tilde{X}^{(1)}, \ldots, \tilde{X}^{(p)}] \in \mathbb{R}^{n \times p}$ of $X = [X^{(1)}, \ldots, X^{(p)}] \in \mathbb{R}^{n \times p}$ is constructed so that, for any subset $S \subseteq [p]$,

$$(X, \tilde{X})_{\text{swap}(S)} \stackrel{d}{=} (X, \tilde{X}).$$

where $(X, \tilde{X})_{\text{swap}(S)}$ swaps $X^{(j)}$ and $\tilde{X}^{(j)}$ for each $j \in S$. The knockoff variables are constructed without using $Y$, so that $\tilde{X} \perp\!\!\!\perp Y \mid X$.

We set variable importance measure as coefficients: $Z_k = |\hat{\delta}'_k(\lambda)|$, $\tilde{Z}_k = |\hat{\delta}'_{k+K}(\lambda)|$. Alternatively, we can also use $Z_k = \sup\{\lambda \geq 0 : \hat{\delta}'_k(\lambda) \neq 0\}$, the lambda value where each feature/knockoff enters the lasso path (meaning becomes nonzero). The knockoff filter works by comparing the $Z_k$'s to the $\tilde{Z}_k$'s and selecting only variables that are clearly better than their knockoff copy. The reason why this can be done is that, by construction of the knockoffs, the correction-null statistics are pairwise exchangeable. This means that swapping the $Z_k$ and $\tilde{Z}_k$'s corresponding to null variables leaves the koint distribution of $(Z_1, \ldots, Z_K, \tilde{Z}_1, \ldots, \tilde{Z}_K)$ unchanged. Once the $Z_k$ and $\tilde{Z}_k$'s have been computed, different contrast functions can be used to compare them. In general, we must choose an anti-symmetric function $a$ and we compute the symmetrized knockoff statistics $W_k = a(Z_k, \tilde{Z}_k) = -a(\tilde{Z}_k, Z_k)$ such that $W_k$ indicates that $X_k$ appears to be more important than its own knockoff copy. We use difference of absolute values of coefficients by default, but many other alternatives (like signed maximum) are also possible.

For repeat $b$, let $W_k^{[b]}$ denote the knockoff statistic for coordinate or group $k$. The knockoff+ threshold is

$$T^{[b]} = \min\left\{t \in \mathcal{W}^{[b]} : \frac{1 + \#\{k : W_k^{[b]} \leq -t\}}{\#\{k : W_k^{[b]} \geq t\} \vee 1} \leq q\right\},$$

$$\mathcal{W}^{[b]} = \{|W_k^{[b]}| : |W_k^{[b]}| > 0\}.$$

If this set is empty, we set $\hat{A}^{[b]} = \emptyset$; otherwise $\hat{A}^{[b]} = \{k : W_k^{[b]} \geq T^{[b]}\}$.

# E. Proof of Theorem 3.3: Stability Selection Control

*Proof.* **PFER Control:** For each null feature $k \in A^c$, the per-iteration selection probability satisfies $P(k \in \hat{A}^{[b]}) \leq \alpha$. This per-iteration bound is an assumption of the stability-selection argument; in practice, $\tau$ is the knockoff threshold computed from the knockoff statistics. Define $V_k^{[b]} = \mathbf{1}\{k \in \hat{A}^{[b]}\}$ and assume the repeated knockoff draws are independent conditional on the data, with $\mathbb{E}[V_k^{[b]}] \leq \alpha$. The selection frequency $\hat{\Pi}_k = \frac{1}{B}\sum_{b=1}^{B} V_k^{[b]}$ is an average of independent Bernoulli variables with means bounded by $\alpha$. By Hoeffding's inequality:

$$P\left(\hat{\Pi}_k \geq \pi\right) \leq \exp\left(-2B(\pi - \alpha)^2\right) \quad \forall \pi > \alpha.$$

Summing over all null features and applying linearity of expectation:

$$\mathbb{E}\left[|\hat{A}(\pi) \cap A^c|\right] = \sum_{k \in A^c} P\left(\hat{\Pi}_k \geq \pi\right) \leq |A^c| \exp\left(-2B(\pi - \alpha)^2\right).$$

**FDR Control:** The stability-thresholded set $\hat{A}(\pi)$ is not, by itself, the object for which the derandomized knockoff FDR theorem applies. For FDR control, we use the e-value aggregation procedure of Ren & Barber (2024): each knockoff run produces a valid knockoff statistic and the repeated runs are converted into e-values, after which e-BH returns a selected set $\hat{A}_{\text{eBH}}$ satisfying $\text{FDR}(\hat{A}_{\text{eBH}}) \leq q$ under the Model-X knockoff assumptions. Thus, the PFER display above applies to the stability-thresholded set, while the formal FDR guarantee applies to the e-BH aggregated set. $\square$

We repeat the main experiments using this control, results are below.

| Dataset | Matched (FDR, Power) | Mismatched (FDR, Power) |
|---|---|---|
| | **Sparse** | |
| CovidCom | (0.00, 0.75) | (0.01, 0.74) |
| Diabetes | (0.00, 0.50) | (0.05, 0.21) |
| Support2 | (0.03, 0.92) | (0.05, 0.91) |
| | **Dense** | |
| CovidCom | (0.00, 0.88) | (0.04, 0.75) |
| Diabetes | (0.10, 0.42) | (0.11, 0.36) |
| Support2 | (0.09, 1.00) | (0.15, 0.83) |

# F. Heuristic One-Pass Penalty Calibration for Naive SGShift

Given the assumption of i.i.d. observations and the exponential family distribution to generate the dependent variable $y$, $f(X_i)$ determines the $\mathbb{E}[y_i \mid X_i]$ under domain $S$, and $\delta$ captures the shift.

The negative log-likelihood function of $\delta$ can be written as

$$L(\delta) = \frac{1}{n_T} \sum_{i=1}^{n_T} \left\{ \psi\big(f(X_i) + \phi_i^\top \delta\big) - y_i\big(f(X_i) + \phi_i^\top \delta\big) \right\} = \ell\big(f(X_T) + \phi_T^\top \delta, y_T\big)$$

where $\psi(\cdot)$ is uniquely determined by the link $g(\cdot)$, $\ell\big(\eta, y\big) = \psi\big(\eta\big) - y\eta$ where $\eta = f(X) + \phi^\top \delta$.

We regularize the GAM loss with an $\ell_1$-penalty

$$\hat{\delta}(\lambda) = \arg \min_{\delta \in \mathbb{R}^K} \left\{ \frac{1}{n_T} \sum_{i=1}^{n_T} \big(\psi(\eta_i) - y_i \eta_i\big) + \lambda \|\delta\|_1 \right\}$$

The score vector is

$$\nabla L(\delta) = \frac{1}{n_T} \sum_{i=1}^{n_T} \left[ \psi'\big(f(X_i) + \phi_i^\top \delta\big) - y_i \right] \phi_i$$

Evaluated at $\delta = 0$, $\gamma := \nabla L(\delta)\big|_{\delta=0} = \frac{1}{n_T} \sum_{i=1}^{n_T} \left[ \psi'\big(f(X_i)\big) - y_i \right] \phi_i$, where $\psi'(\cdot)$ is the canonical mean function.

For a general correlated design, the exact KKT condition is

$$0 \in \nabla L(\hat{\delta}) + \lambda \partial \|\hat{\delta}\|_1,$$

so selection depends on the gradient at $\hat{\delta}$, not only on $\nabla L(0)$. The following calculation should therefore be interpreted as a one-pass orthogonal-design heuristic for choosing a penalty scale.

Assume each row $\phi_i$ is sub-Gaussian with i.i.d. coordinates and that every coordinate of $\phi_i$ and $y_i$ has been centered and variance-normalized.

Let $\sigma_\gamma^2 := \mathbb{V}(y_i \mid X_i) = \psi''\big(f(X_i)\big)$, where $\psi''(\cdot)$ is the variance function of the canonical exponential-family model.

Let the true parameter be $\delta^* \in \mathbb{R}^K$ with support $A \subseteq [K]$, $|A| = a$, so $\delta_j^* = 0$ for $j \in A^c$.

**Null coordinates.** Let $j$ be a null coordinate $j \in A^c$ among $K - a$ null coordinates.

At the true parameter $\delta^*$, null-score centering gives

$$\mathbb{E}\left[ Y_i - \psi'\{f(X_i) + \phi_i^\top \delta^*\} \mid X_i \right] = 0,$$

and hence

$$\mathbb{E}\left[ \big(Y_i - \psi'\{f(X_i) + \phi_i^\top \delta^*\}\big) \phi_{ij} \right] = 0.$$

The following one-pass calculation approximates this null score by the score at $\delta = 0$.

Under this one-pass approximation, define $Z_{ij} = \big(y_i - \psi'(f(X_i))\big)\phi_{ij}$, we treat $\{Z_{ij}\}_{i=1}^{n_T}$ as i.i.d., mean-zero, and sub-Gaussian for the purpose of the heuristic calculation.

$$\mathbb{V}[Z_{ij}] = \mathbb{E}\big[\big(y_i - \psi'(f(X_i))\big)^2 \phi_{ij}^2\big] = \mathbb{E}\left[ \psi''(f(X_i))\phi_{ij}^2 \right] = \mathbb{E}\left[ \psi''(f(X_i)) \right] = \sigma_\gamma^2$$

where the second equality follows under the additional orthogonal-design and homoskedastic working approximation used for this heuristic.

Under mild moment conditions, the Lindeberg-Feller central limit theorem (CLT) implies

$$\frac{1}{\sqrt{n_T}} \sum_i \big\{ \big(y_i - \psi'(f(X_i))\big)\phi_{ij} \big\} \xrightarrow{d} N\big(0, \sigma_\gamma^2\big) \quad \text{if } j \in A^c$$

Therefore, for null coordinates, we have

$$\sqrt{n_T}\,\gamma_j = -\frac{1}{\sqrt{n_T}}\sum_{i=1}^{n_T}\big(y_i - \psi'(f(X_i))\big)\phi_{ij} \xrightarrow{d} N(0,\sigma_\gamma^2) \quad \text{if } j \in A^c.$$

**False-selection probability and plug-in mFDR estimate.** Under the same orthogonal one-pass heuristic, we approximate selection by the event $|\gamma_j| > \lambda$, so the null-coordinate error rate is

$$Pr(j \text{ selected} \mid j \in A^c) = Pr(|\gamma_j| > \lambda) = 2\Big\{1 - \Phi\big(\frac{\lambda\sqrt{n_T}}{\sigma_\gamma}\big)\Big\}$$

where $\Phi$ is the standard normal CDF.

Following (Miller & Breheny, 2019), the marginal FDR is

$$\text{mFDR}(\lambda) = \frac{\mathbb{E}[\#\text{False Discoveries}]}{\mathbb{E}[\#\text{Selected}]}$$

Plugging in the null probability above yields

$$\widehat{\text{FDR}}(\lambda) = \min\Big\{\frac{2K\big(1 - \Phi\big(\lambda\sqrt{n_T}/\sigma_\gamma\big)\big)}{|\hat{A}(\lambda)| \vee 1}, 1\Big\},$$

where we use $K$ rather than the unknown number of null coordinates.

A practical one-pass rule that targets an approximate marginal-FDR level $\alpha$ under the orthogonal-design heuristic is

$$\hat{\lambda}_\alpha = \min\Big\{\lambda : \widehat{\text{FDR}}(\lambda) \leq \alpha\Big\}$$

# G. Datasets

All datasets are listed as below, and the full preprocessing code from raw data, together with the preprocessed data, are available in the source code, except restricted access COVID-19 Hospitalization data where we provide detailed fetching code and data version information from NIH All of Us Research Program (of Us Research Program Investigators, 2019). Standardization is performed within the pipeline to ensure that features with larger values don't disproportionately influence the $\ell_1$ regularization penalty.

|  | Diabetes readmission | COVID-19 | SUPPORT2 |
|---|---|---|---|
| Total samples | 73,615 | 16,187 | 9,105 |
| Features | 33 | 30 | 64 |
| Source size | 49,213 | 11,268 | 5,453 |
| Target size | 24,402 | 2,219 | 1,817 |
| Domain split | Emergency room admission | New variant | Death in hospital |

*Table 2.* Dataset summary.

**Diabetes 30-Day Readmission**   The Diabetes 130-US Hospitals dataset, available through the UCI Machine Learning Repository (https://archive.ics.uci.edu/dataset/296/diabetes+130-us+hospitals+for+years+1999-2008), comprises 101,766 encounters of diabetic patients across 130 U.S. hospitals between 1999-2008 (Strack et al., 2014). We fetch the data following TableShift's procedure (Gardner et al., 2023).

We define the source domain as 49,213 non-ER admissions (elective or urgent) with 25,196 readmitted patients, and the target domain as 24,402 ER admissions with 10,684 readmitted patients, with the binary classification task being prediction of 30-day readmission risk.

**COVID-19 Hospitalization**   The COVID-19 cohort is part of the NIH All of Us Research Program (of Us Research Program Investigators, 2019), a (restricted access) dataset containing electronic health records for 16,187 patients diagnosed with COVID-19 between 2020-2022. Features include demographic variables (age, gender, race), temporal indicators (diagnosis date relative to Omicron variant emergence), comorbidity status for 13 chronic conditions (diabetes, COPD), and diagnostic context (EHR vs. claims-based). We partition the data into three temporal groups: a source domain of 11,268 patients diagnosed prior to the beginning of 2022 with 2,541 patients hospitalized, a target domain of 2,219 patients diagnosed in January 2022 (early Omicron era) with 359 patients hospitalized. The binary classification task predicts hospitalization status (inpatient vs. outpatient).

**SUPPORT2 Hospital Charges**   From the Study to Understand Prognoses Preferences Outcomes and Risks of Treatment (SUPPORT2), publicly available via the UCI repository (https://archive.ics.uci.edu/dataset/880/support2) containing 9,105 critically ill patients (Connors et al., 1995).

The source domain is specified as 5,453 patients who survived hospitalization and the target domain as 1,817 in-hospital deaths. The regression task is defined as a prediction of $\log_{10}$(total hospital costs per patient).

# H. Model Hyperparameters

We used standard implementations of classical machine learning models from `scikit-learn`, with hyperparameters either set to commonly used defaults or manually tuned for stability and performance. Supplementary table 3 summarizes the key hyperparameters for each model. Unless otherwise stated, all models were trained using their default solver settings. Random seeds were fixed via `random_state` to ensure reproducibility.

| Model | Hyperparameters |
|-------|-----------------|
| Decision Tree (Classifier) | `max_depth=4, random_state={seed}` |
| Support Vector Machine (Classifier) | `kernel='rbf', C=1.0, probability=True, random_state={seed}` |
| Gradient Boosting Classifier | `n_estimators=100, random_state={seed}` |
| Logistic Regression | `max_iter=200, random_state={seed}` |
| Decision Tree (Regressor) | `max_depth=4, random_state={seed}` |
| Support Vector Machine (Regressor) | `kernel='rbf', C=1.0` |
| Linear Regression | default settings |
| Gradient Boosting Regressor | `n_estimators=100, random_state={seed}` |

*Table 3.* Hyperparameters used for each model. The same random seed (`{seed}`) was applied across models where applicable to ensure consistency.

## I. Sample size

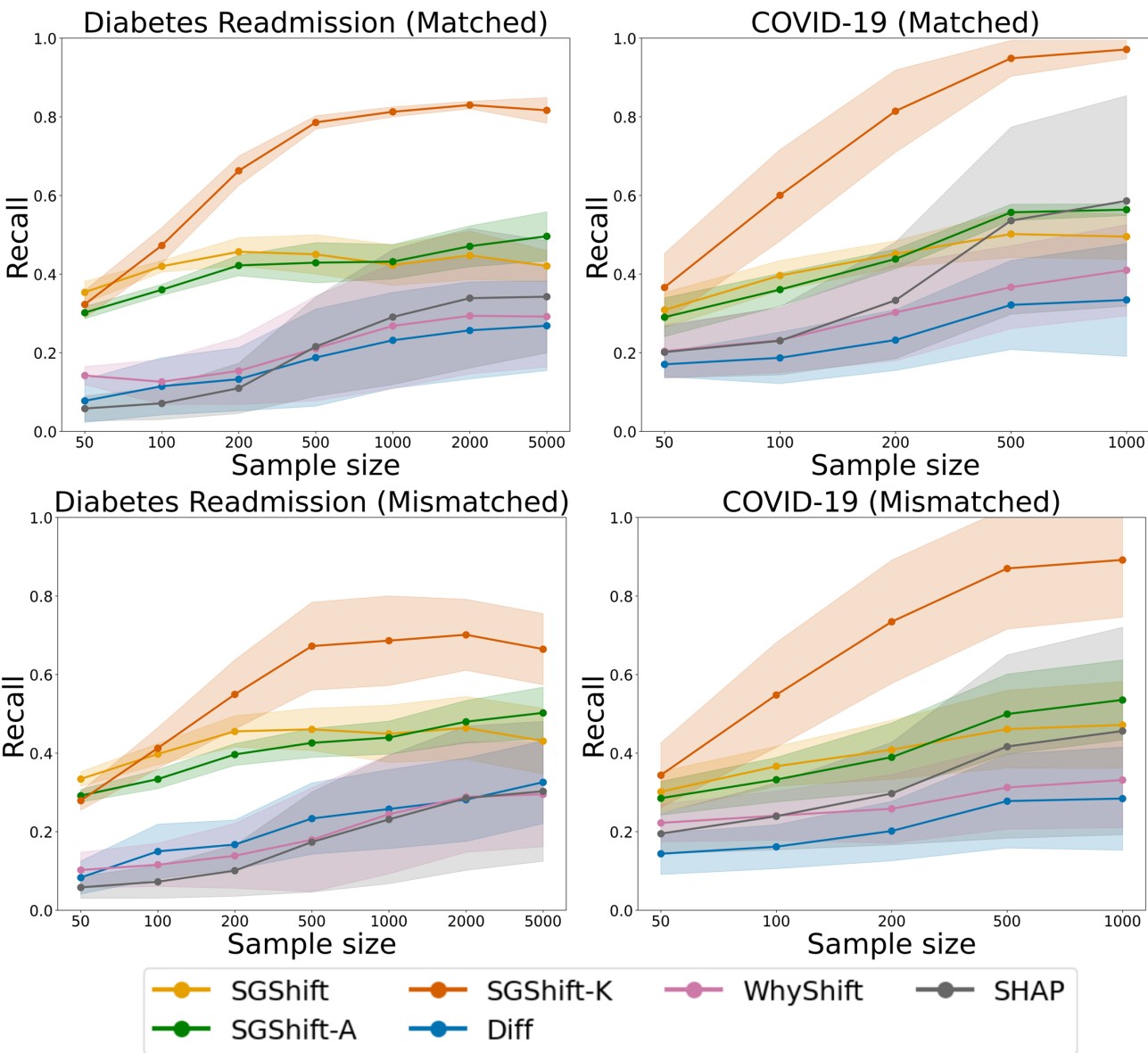

*Figure 4.* **Performance across sample sizes.** Sample size is varied from 50 to 1000. 95% CI's are shown across 16 simulation settings. Recall is measured at fixed FPR 5%.

## J. Global calibration shifts

**Global shifts.** It may be true that all features are shifted in the same direction in a new domain due to differences in sensor calibration. In this case identifying specific shifted features may be difficult as all are perturbed slightly. We simulate this global effect where only a few true features having a conditional shift, and the rest are perturbed by noise with absolute values from 0.01 to 0.3 while the absolute values of true shifts are 3. Results are reported in Figure 5. SGShift-K still strongly identifies true shifted features with AUC $> 0.9$, even when all features are shifted slightly, and individual features are not over or under prioritized. Interestingly, for all methods, performance is relatively unchanged as the scale of the background shift increases. This may be due to the intercept term accumulating the background shift, as opposed to attributing it to any individual feature.

*Figure 5.* **Global calibration shift performance.** Performance as a background conditional shift is increased in scale. X-axis represents strength of the background shift, as 0.01x-0.3x the true shift magnitude.

# K. Interaction shifts

**Interaction shifts.** We assess each method's performance in identifying features which shift in the interaction space in the SUPPORT2 dataset, where features are continuous. The goal is to detect individual features contributing to shift through interactions with other features. We consider two cases of SGShift-K, underspecified, where the basis function does not include interaction terms, and overspecified, where SGShift contains both second and third tier interactions. Results are reported in Table 4. In both cases, regardless of how the basis function is specified, SGShift displays strong performance.

|  | Diff | WhyShift | SHAP | SGShift-K – underspecified | SGShift-K – overspecified |
|---|---|---|---|---|---|
| Matched | 0.75 | 0.73 | 0.80 | **0.92** | 0.90 |
| Mismatched | 0.72 | 0.73 | 0.78 | **0.89** | 0.87 |

*Table 4.* **Performance in detecting interaction shifts.**

