# OpenReview forum: "Explaining Concept Shift with Interpretable Feature Attribution"
_ICML.cc/2026/Conference — ICML 2026 regular_

### Official Review · Reviewer_KRA1 · 2026-03-10

**Soundness:** 3
**Presentation:** 3
**Significance:** 3
**Originality:** 3
**Overall Recommendation:** 5
**Confidence:** 4

**Summary:**

The paper studies concept shift in tabular data and proposes a procedure to identify a sparse set of features whose relationship with the label appears to change between source and target domains. It develops three variants: (1) SGShift, a basic l1-regularized additive correction model; (2) SGShift-A to separate source-model misspecification from true shift; and (3) SGShift-K, which combines the approach with knockoffs to reduce false discoveries. Empirically, the paper reports improved ROC AUC on semi-synthetic benchmarks across a range of settings and presents real-data case studies to illustrate the plausibility of the selected features.

**Compliance With Llm Reviewing Policy:**

Affirmed.

**Final Justification:**

The authors addressed most of my concerns and even though some are partially resolved, I think the paper satisfies publication requirements.

**Key Questions For Authors:**

Please see previous section

**Limitations:**

The paper uses the generic impact statement. See previous section for limitations and suggestions for improvement.

**Strengths And Weaknesses:**

The paper addresses a relevant and important problem using an intuitive approach that is written well and easy to follow. The proposed method is empirically promising, with strong results on the reported semi-synthetic benchmarks, and the paper makes a meaningful effort to combine theoretical and practical considerations in developing the approach. The paper is generally well written, although some sentences could be restructured or rephrased for greater rigor and clarity (see minor comments below).

Despite these strengths, there are several major concerns, along with a number of minor ones. Below is a list in no particular order of importance by category.

Major comments:

1. There is a slight misalignment between the paper’s main claims and its evaluation protocol. The experiments evaluate methods primarily using ROC AUC and recall at a fixed 5\% FPR, which assess the quality of a feature ranking. However, the paper’s central claim is not to rank features, but to identify a sparse support set $A$ of shifted features. That is closer to a selection problem than a pure ranking problem. Moreover, when $A$ is sparse, as assumed in much of the paper’s motivation and simulations, the positive class is rare. This can make ROC AUC appear strong even when precision is modest. The evaluation would be more aligned with the paper’s claims if it also reported selection-oriented metrics such as precision@K, recall@K, F1 at a selected K, PR AUC, or empirical FDR/precision for SGShift-K. These metrics would better support claims such as “SGShift can identify shifted features much more accurately than baseline methods,” especially given the paper’s emphasis on sparse recovery and reduced false discovery.

2. The considered baselines are reasonable but not fully comprehensive. They are adapted versions of existing methods, rather than the closest prior approaches designed to explain performance differences under concept shift. This is partly understandable, since the paper formulates the problem in a somewhat different way than prior work. However, the current comparison set still feels incomplete. In particular, methods such as Singh et al. (2024), and possibly Chen et al. (2022) in an assumption-matched setting, seem close enough to warrant either numerical comparison or a more explicit discussion of why such a comparison would be unfair. While these methods do require additional structural choices, such as variable decomposition or stronger identification assumptions, those differences do not by themselves make comparison impossible. At minimum, the paper should better justify why the current baselines are the most relevant ones for its claims.

3. The semi-synthetic construction perturbs $g(E[Y ∣ X])$ on selected features after fitting a generator model on the source data. That setup is reasonable, but it is also structurally close to the paper’s own additive sparse-correction view of shift, and may therefore be somewhat favorable to SGShift. The additional experiments on dense shifts, interaction/global shifts, feature correlation, and high-dimensional settings help mitigate this concern to a degree, but the main empirical story is still driven by a semi-synthetic design that is congenial to the method. The paper would be stronger if it included at least one benchmark in which the target-domain shift mechanism is generated in a way that is less directly aligned with SGShift’s modeling assumptions, or explained more clearly why the current simulation design is not unduly favorable.

4. The claim that "real-world concept shift is sparse" is not fully established by the evidence presented. What the paper shows is that SGShift can recover most of the lost target performance using relatively few features. However, this is not the same as showing that the underlying concept shift itself is sparse in the data-generating process. It could also reflect the inductive bias of the correction model, the effect of regularization, redundancy among covariates, or the existence of a small high-leverage repair set rather than true sparse shift. The paper should tone down this claim and present the real-data evidence more carefully as showing that concept shift can often be repaired or approximated well using a small number of features.

5. The matched and mismatched settings of the numerical results should be better motivated and explained. It would help the reader if the paper explained more explicitly why this distinction matters, what type of robustness the mismatched setting is intended to evaluate, and how it relates to the practical use case of SGShift. As written, the experimental design is understandable, but its rationale is under-explained.

6. The paper’s theory is ambitious, but several arguments in the proofs are either under-specified or not fully matched to the claims in the main text. The theoretical claims should either be stated more carefully or supported with more rigorous proofs. Below is an example list.

6.1. Section 3.4 motivates Theorem 3.1 as requiring assumptions mainly on the between-domain difference $\Delta$, rather than on the full target regression function. However, the proof relies on a more standard high-dimensional penalized-GLM regime than what the main text suggests. This includes the sub-Gaussian design assumptions, differentiability/curvature conditions on the loss, and an RSC argument deferred to Appendix A. These assumptions may be reasonable if made explicit, but they are stronger and more global than the theorem discussion currently suggests. In addition, Appendix A states an RSC-style inequality in a form that appears stronger than the local or cone-restricted versions more commonly used in high-dimensional GLM analyses.

6.2. The normalization of the loss and the scaling of $\lambda$ appear inconsistent. Equation (2) defines the estimator as $\hat{\delta} = \arg\min_{\delta \in \mathbb{R}^K} \lbrace L(\delta) + \lambda ||delta|| \rbrace$ while the proof writes the loss explicitly as a \emph{sum} over target samples $\hat{\delta}(\lambda) = \arg\min_{\delta} \lbrace \sum\limits_{i=1}^{n_T} \bigl(\psi(\eta_i) - y_i \eta_i\bigr) + \lambda \|\delta\|_1 \rbrace$. However, Theorem 3.1 states a scaling equivalent to $\lambda \asymp \sqrt{\frac{\log K}{n_T}}$, which is the standard order when the loss is \emph{averaged}, not summed. If the objective uses the summed loss, the corresponding penalty scale should be different. This makes it unclear which normalization is intended, since the subgradient and error-bound calculations depend directly on that choice.

6.3. Theorem 3.1 establishes an $\ell_2$ error bound of the form $ |\hat{\delta} - \delta^\ast|_2^2 \lesssim \frac{a \log K}{n_T}$, but this is not the same as support recovery. The theorem, as stated, does not justify the strong support-recovery claim.

6.4. There is no corresponding theorem for SGShift-A, despite misspecification being a central motivation of the paper.

6.5. Appendix E states $\hat{\delta}_j (\lambda) \neq 0$ if and only if $ |\gamma_j| > \lambda $, where $\gamma = \nabla L(\delta)$ with $\delta=0$. This type of thresholding statement is generally valid only in orthogonal or decoupled settings, not for a general correlated-design penalized GLM. As written, this is not a correct general KKT characterization.

Minor comments:

Below is a list of minor issues that should either be addressed or sufficiently justified. I would be happy if the authors retain their original choices, provided they explain more clearly why those choices are preferable.

1. There are several inaccurate claims or writing issues in the abstract:

- The first sentence of the abstract is not entirely accurate. Concept shift does not mean the ML model has "learned a fundamentally incorrect representation". It means the model does not extend well to (may become miscalibrated or suboptimal on) the target domain.

- The second sentence of the abstract, the phrase "these shifted features" is introduced before such features have been clearly defined. As written, the phrase has no explicit antecedent in the preceding sentence.

- Similarly, "how one dataset differs from another" is too broad and should be tightened for rigor.

- SGShift abbreviation is never specified in the paper. It would help to clarify what SG stands for.

- The abstract describes SGShift as ``a model for detecting concept shift in tabular data''. That wording suggests the method can determine whether concept shift has occurred. However, the paper, as developed, is more focused on identifying a sparse set of features associated with concept shift, implicitly assuming that concept shift is already present. This claim should therefore either be toned down, or supported more directly by showing that SGShift does not select features when concept shift is absent.

- "This framework enables SGShift to adapt powerful statistical ...", is not entirely accurate. Those tools are not just external adaptations that the framework permits. They are also the concrete procedures used to build the best SGShift variants.

- "extensive experiments in" -> "extensive experiments on".

2. In the introduction, the paper should elaborate on "However, in other cases, it may be necessary to fix issues in an an underlying data pipeline". The examples the paper gives at the end of the same paragraph are sufficient, but slightly disconnected by sentence structure. A restructure of that paragraph would benefit flow.

3. The current lowercase notation of $y$ in expressions such as $P(y|X)$ is not ideal. A cleaner and more standard framing of concept shift at the distribution level is $P(Y|X)$, since lowercase $y$ usually denotes a realized value of the label rather than the full conditional distribution that the paper is discussing. If the paper intends a pointwise statement, then notation such as $P(Y=y \mid X=x)$ or $p(y \mid x)$ would be more precise.

4. Section 1.1., the first sentence "shifts in the marginal feature distribution, $P(X)$, e.g. covariate shift ..." is not entirely correct. A cleaner definition would be "shifts in the marginal feature distribution, $P(X)$, while assuming that $P(Y∣X)$ remains unchanged, i.e., covariate shift.". The "e.g." suggests covariate shift is just one example of marginal-feature shifts, when in standard usage it is essentially the case defined by changing $P(X)$ while keeping $P(Y∣X)$ fixed.

5. There are many "citep" used in places where "citet" should be used in the related works. There are also several typos in quotation marks. Please correct throughout the paper.

6. The comparison to Singh et al. (2025) should be discussed in more detail. As written, the paper notes only that Singh et al. (2025) discovers subgroups within the data for which to produce feature-level explanations. Given how close that sounds to the present goal, the paper should explain more explicitly why this framework is not directly applicable here. My understanding is that Singh et al. (2025) targets heterogeneous subgroup-specific performance decay via hierarchical hypothesis testing, whereas the present paper aims to recover a global sparse support set of features associated with conditional shift. If that is the intended distinction, it should be stated clearly.

7. The sentence “A shift happens such that $P_T(y \mid X) \neq P_S(y \mid X)$ for at least one feature in $X$” is conceptually imprecise. The inequality $P_T(y \mid X) \neq P_S(y \mid X)$ is a statement about the full conditional distribution given the entire feature vector $X$, not about an individual feature. If the paper wishes to move from concept shift at the distribution level to feature-level attribution, then the contribution of individual covariates should be defined more carefully. Since this is the central problem of the paper, that transition deserves more precise treatment.

8. The sentence “For any given training point $X$, we see either a label $y$ from distribution $S$ or distribution $T$, but never both” is not entirely precise. It treats the same feature vector $X$ as if it could have been observed under both domains and assigned two labels. That is usually not the setup in concept shift. Even if the same covariate value $x$ appears in both domains, concept shift means the conditional distributions $P_S(Y\mid X=x)$ and $P_T(Y\mid X=x)$ may differ; it does not mean there is a single training point whose label is revealed in either the source or target domain. A clearer wording would be that we observe labeled samples from the source and target domains separately, but not paired labels for the same realized sample under both domains.

9. The introduction of knockoffs in the SGShift framework could be motivated more clearly. Before Section 3.3, the paper mentions knockoffs as one of the tools used by SGShift, but does not yet explain how they enter the procedure or why they are needed in this setting. More specifically, the paper should better motivate why the baseline SGShift feature-selection step is prone to false discoveries, and why knockoffs are the appropriate remedy here. Section 3.3 explains the mechanics of the knockoff construction, but the motivation for why false discovery is a central concern in SGShift remains underdeveloped.

10. The sentence "despite the fact that the outcome we are attempting to recovery the sparsity pattern for is never directly observed." can be better worded. Also, the entire "We show ... set, despite the fact" sentence should be separated into two sentenced for better flow.

11. "However, other sparse regression methods can be applied out-of-the-box to fit the characteristics of specific data distributions." It is not entirely clear what this sentence means. Perhaps discussing this in more detail, with examples, would aid the reader.

12. "In order to control the sparsity level ... solve" -> solves.

13. The sentence "The main absorption idea is that the error from fitting occurs in both domains, while the conditional shift occurs only in the target domain" is not entirely accurate. A source model’s misspecification may manifest differently in source and target distributions because the covariate distribution changes, feature interactions differ in prevalence, noise levels differ, or the model extrapolates differently in the target region. So the absorption idea of the paper is really an approximation/modeling assumption: the part of the error due to generic model misspecification is assumed to be sufficiently shared across domains that a common correction term can soak it up, while the residual target-only part is treated as shift. The paper should avoid over-claiming here and clearly state this as an assumption.

14. "Knockoffs generate synthetic features that mimic the correlation structure of the real data to limit false discoveries". A more precise statement is: Knockoffs generate synthetic features that preserve the dependence structure of the original features while being conditionally unrelated to the response $Y$, enabling controlled variable selection and limiting false discoveries.

15. Figures could improve by using different markers per method, rather than the same circle marker across all methods. The figures should be readable without relying on color alone. A,B,C,D subfigures are not specified in Figure 3.

---

> ### Author Rebuttal · Authors · 2026-03-29
>
> Thank you for the accurate summary of our method and for highlighting SGShift’s innovation compared to prior work. The reviewer has raised 10 major comments and 15 minor comments. We summarize each review comment and provide a point-by-point response below.
>
> Please let us know if there are any additional questions or requests for further clarification. If the reviewer feels that their concerns have been adequately addressed, we would be grateful if they would consider updating their evaluation.
>
> Due to the extent of the feedback, we are unable to fit the response to all comments in the initial rebuttal but will respond to the first 5 points and upload the rest once rebuttals are released.
>
> > There is a slight misalignment between the paper’s main claims and its evaluation protocol...
>
> KRA1-W1: Metrics
>
> Thank you for the comment. We reported AUC and recall at fixed FPR because the baselines and other SGShift variants produce rankings and cannot target a specified FDR. For SGShift-K, we additionally computed empirical FDR and power at nominal FDR below 10% following Appendix D. FDR is controlled in all settings except dense mismatched Support2, with generally strong power. We will include this analysis in the camera-ready.
>
> **Sparse**
>
> |Dataset|Matched(FDR,Power)|Mismatched(FDR,Power)|
> |---|---|---|
> |CovidCom|(0.00,0.75)|(0.01,0.74)|
> |Diabetes|(0.00,0.50)|(0.05,0.21)|
> |Support2|(0.03,0.92)|(0.05,0.91)|
>
> **Dense**
>
> |Dataset|Matched(FDR,Power)|Mismatched(FDR,Power)|
> |---|---|---|
> |CovidCom|(0.00,0.88)|(0.04,0.75)|
> |Diabetes|(0.10,0.42)|(0.11,0.36)|
> |Support2|(0.09,1.00)|(0.15,0.83)|
>
>
> > The considered baselines are reasonable but not fully comprehensive...
>
> KRA1-W2: Baselines
>
> Thank you for the suggestion. Two further baseline classes are: (1) methods without target labels, e.g. Chen et al. (2022), and (2) methods requiring dataset-specific prior knowledge, e.g. Singh et al. (2024).
>
> For (1), methods without target labels cannot identify which features shifted when only $Y_T$ is perturbed. In our simulations, $Y_T$ changes for the same $X_T$, with only natural covariate shift, so Chen et al. selects the same features regardless of the true shifted set; we confirmed this in several simulations and therefore did not include this class.
>
> For (2), these methods require prior information unavailable in our setting, rather than assumption mismatch. Singh et al., for example, needs variables partitioned into unshifted $W$ and conditional covariates $Z$, specified via a causal graph or invariant features. No method in our experiments has access to this information, and their code does not support the simplified choice $Z=X,W=\emptyset$. We will clarify in the camera-ready that all compared methods are evaluated without dataset-specific prior information.
>
>
> > The semi-synthetic construction perturbs  on selected features after fitting a generator model on the source data...
>
> KRA1-W3: Simulations
>
> Thank you for the comment. Our semi-synthetic design is a general representation of conditional distribution shift, where selected features have perturbed relationships with the target. It does not encode method-specific assumptions and is therefore not designed to favor any method. By contrast, an unfair simulation would build a method’s assumptions directly into the data-generating process; we do not do this. We will revise the exposition to better explain this choice and why it is not method-favorable.
>
> > The claim that "real-world concept shift is sparse" is not fully established by the evidence presented...
>
> KRA1-W4: Interpretation
>
> Please see our response to jwpj-W1. We will emphasize SGShift's results are not meant to be interpreted in a causal manner but in the ML model sense.
>
> > The matched and mismatched settings of the numerical results should be better motivated and explained...
>
> KRA1-W5: Matched vs mismatched
>
> Thank you for the comment. This setup is meant to show that the target model need not be retrained in the same way as the base model, which is realistic when the target domain has fewer samples, different ML frameworks are used, or some model misfit is present. We will clarify this motivation in the camera-ready.

---

> > ### Author Rebuttal · Reviewer_KRA1 · 2026-04-01
> >
> > First, I want to note (particularly to the chairs) that the length of my review is not meant to suggest that I see the paper as having an unusual number of problems. I enjoyed reading the paper and found the core idea interesting. I wrote a detailed review because I wanted to be as helpful as possible and provide concrete suggestions that could strengthen the work.
> >
> > On the rebuttal, I appreciate the additional clarification on several points, especially the motivation for the matched versus mismatched settings and the explanation for why some prior methods were not included as baselines. The added empirical FDR/power analysis for SGShift-K is also helpful and partially addresses my concern that the evaluation should better align with the paper’s sparse-selection and false-discovery claims.
> >
> > That said, based on the current rebuttal, I do not think my main concerns are fully resolved yet.
> >
> > Evaluation: the added FDR/power results are useful, but they only address SGShift-K. My broader concern was that the paper’s central claim is about identifying a sparse support set of shifted features, whereas the main evaluation still relies primarily on ranking-oriented metrics such as ROC AUC and recall at fixed FPR. I still think the paper would benefit from more selection-oriented metrics or a clearer reframing of the empirical claims.
> >
> > Simulation: I do not think the simulation concern is fully addressed. My point is not that the semi-synthetic design hard-codes the proposed method, but that it remains structurally congenial to the paper’s additive sparse-correction formulation. The rebuttal states that the simulation is a general representation of conditional shift, but does not yet explain why this design would not still be somewhat favorable to SGShift relative to less aligned alternatives.
> >
> > Novelty claims: I do not think the rebuttal resolves my concern about the claim that “real-world concept shift is sparse.” My concern was not mainly about causal interpretation, but about the distinction between showing that a small set of features can recover most of the lost performance under SGShift, versus showing that the underlying concept shift itself is sparse in the data-generating process. I still think the paper should tone down that claim.
> >
> > Theory: My largest remaining concern is that the rebuttal so far does not address the theory issues, which were a substantial part of my soundness assessment, including the assumptions behind Theorem 3.1, the loss-normalization issue, the distinction between estimation error and support recovery, the lack of theory for SGShift-A, and the Appendix E KKT claim. Until those points are addressed, I do not think I have enough basis to revise my soundness score.
> >
> > Overall, I appreciate the thoughtful engagement, and some concerns have been partially addressed. However, based on the current rebuttal alone, I do not think the response is yet sufficient to change my overall evaluation. I would be open to revisiting this if the remaining comments, especially the theory-related ones, are addressed clearly in the follow-up response.

---

> > > ### Author Response · Authors · 2026-04-01
> > >
> > > Thank you for carefully reading our response. We sincerely appreciate your constructive feedback, which has helped us strengthen the paper.
> > >
> > > Due to space, we cannot address each minor comment individually, but we broadly agree with them and will incorporate the suggestions in the camera-ready.
> > >
> > > Here is our response to the remaining theoretical questions.
> > >
> > > >6.1. Section 3.4 motivates Theorem 3.1...
> > >
> > > We thank the reviewer; these comments are helpful and we agree the current theory section should be sharpened. Our intended result is an offset-GLM sparse M-estimation theorem: $h_S$ is treated as a fixed offset, and sparsity is imposed only on the target correction $\delta$ in Eq. (1)–(2). However, proving consistency still requires the usual high-dimensional assumptions on the target design and loss. We will therefore revise Theorem 3.1 to use the averaged loss $\bar L_n(\delta)=n_T^{-1}\sum_i\ell_i(\delta)$, state $\lambda\asymp\sqrt{\log K/n_T}$, and replace the current global RSC statement by a standard local/cone-restricted RSC condition. Under $\lambda\ge 2\lVert\nabla\bar L_n(\delta^\*)\rVert_\infty$, decomposability gives the cone condition, and local RSC yields $\lVert\hat\delta-\delta^\*\rVert_2^2\lesssim a\lambda^2\lesssim a\log K/n_T$. This fixes the normalization issue and aligns the proof with standard high-dimensional GLM theory. Eq. (2), Theorem 3.1, and Appendix A/B will be revised in the camera-ready.
> > >
> > > >6.2. The normalization of the loss...
> > >
> > > See above. We will make the normalization explicit by writing the objective with the averaged loss $\bar L_n(\delta)=n_T^{-1}\sum_i\ell_i(\delta)$ throughout, so that the stated scaling $\lambda\asymp\sqrt{\log K/n_T}$ matches the estimator and proof.
> > >
> > > >6.3. Theorem 3.1 establishes an $\ell_2$...
> > >
> > > We agree Theorem 3.1 as stated is not an exact support-recovery theorem. We will revise this claim: the current theorem proves estimation consistency. We will also revise the surrounding text so it does not suggest exact support recovery from Theorem 3.1; exact support recovery would require additional assumptions which are not part of the current theorem. The paper’s formal feature-selection guarantee is Theorem 3.2 for SGShift-K, which gives false-discovery control rather than exact support recovery.
> > >
> > > >6.4. There is no corresponding theorem for SGShift-A...
> > >
> > > We agree a parallel theorem can be helpful. Eq. (3) can be rewritten as a weighted $\ell_1$-penalized offset-GLM on the augmented parameter $\beta=(\omega,\delta)$ with augmented rows $z_i^{(S)}=(\phi_i^{(S)},0)$ and $z_i^{(T)}=(\phi_i^{(T)},\phi_i^{(T)})$. Under the corresponding score bounds and a local/cone-restricted RSC condition for this augmented design, the same argument yields $\lVert\hat\omega-\omega^\*\rVert_2^2+\lVert\hat\delta-\delta^\*\rVert_2^2\lesssim s_\omega\lambda_\omega^2+s_\delta\lambda_\delta^2$. We will add this theorem in the camera-ready.
> > >
> > > >6.5. Appendix E states $\hat{\delta}_j(\lambda)\neq 0$...
> > >
> > > We agree Appendix E does not currently state a correct general KKT thresholding rule. For a correlated-design penalized GLM, the exact condition is $0\in\nabla\bar L_n(\hat\delta)+\lambda\partial\lVert\hat\delta\rVert_1$, i.e. selection depends on the gradient at $\hat\delta$, not on $\nabla L(0)$. We will label the Appendix E argument as heuristic-only in the camera-ready, and we note the paper’s main selection guarantee in SGShift-K does not rely on it.
> > >
> > > >Evaluation: the added FDR/power...
> > >
> > > We agree that reporting FDR/power for other methods could in principle be informative, but fair comparison is difficult because these methods do not target FDR control and can yield very different selected sets under different penalties or thresholds. For example, when comparing (FDR, power) = (0.05, 0.5) vs (0.2, 0.9), it is unclear which method is better. This is why we evaluate with AUC and fixed FPR instead. We agree these metrics are not fully aligned with accurate feature selection, but they allow fairer comparison across methods with difficult-to-match operating points.
> > >
> > > >Simulation: I do not think...
> > >
> > > The additive form is not a special restriction of the simulation: once a source conditional is fixed, any target conditional can always be written on the chosen model scale as a source term plus a residual correction, i.e., $g\left(\mathbb{E}_T[Y\mid X]\right) = g\left(\mathbb{E}_S[Y\mid X]\right)+\Delta(X)$ for an appropriate residual function $\Delta$. In that sense, the “additive correction” is a reparameterization of conditional shift, on the modeling scale used by SGShift, not a data-generating assumption tailored to SGShift.
> > >
> > > >Novelty claims: I do not think...
> > >
> > > Yes, this is our intended interpretation. Our claim is not that the underlying data-generating or causal process is literally sparse; rather that real-world concept shift can often be well-approximated by a sparse correction in the predictive model class used by SGShift. We will make this clear throughout the camera ready.

---

### Official Review · Reviewer_jwpj · 2026-03-11

**Soundness:** 3
**Presentation:** 3
**Significance:** 3
**Originality:** 3
**Overall Recommendation:** 4
**Confidence:** 3

**Summary:**

This paper studies the problem of explaining concept shift in tabular prediction tasks, changes between source and target domains. The proposed method, SGShift, learns a sparse correction on top of a source-domain predictor, so that identifying shifted features becomes a structured feature-selection problem. The paper also introduces SGShift-A to absorb source-model misspecification and SGShift-K to provide false-discovery control via knockoffs. Empirically, the paper evaluates the method on semi-synthetic settings and real case studies in healthcare and genetics, and the revised version expands the simulations and clarifies the roles of the method variants.

**Compliance With Llm Reviewing Policy:**

Affirmed.

**Final Justification:**

Since no further replies have been received after the rebuttal acknowledgement, no additional justification is necessary.

**Key Questions For Authors:**

1. Can the real-data claims be reframed more carefully?

2. Can the paper provide a more explicit practical recommendation for when to use each of SGShift, SGShift-A, and SGShift-K?

3. In the real-data experiments, is there a stronger baseline or alternative recovery strategy that could help show that the observed sparsity is not too method-specific?

**Limitations:**

See above.

**Strengths And Weaknesses:**

**Strengths**

The paper addresses a meaningful and practically relevant problem. Much prior work on distribution shift focuses on adaptation or detection, whereas this paper asks a more diagnostic question of which features are responsible for the degradation under concept shift? That framing is interesting and useful. The main methodological idea is clean. Modeling target-domain change as a sparse correction to a fixed source predictor is intuitive, and it gives the method a clear statistical structure. I also think the introduction of SGShift-A and SGShift-K makes sense conceptually, as one addresses model misspecification, the other addresses over-selection and false discoveries.

**Weakness**

My main concern is about the real-world claims. The semi-synthetic experiments are convincing because the ground truth is controlled. In the real case studies, however, it is still hard to know whether the method is isolating true concept shift, as opposed to a mixture of concept shift, measurement drift, label drift, representation mismatch, or dataset-specific confounding. The IPW analysis is helpful, but it only shows that covariate shift correction alone explains little of the performance gap in these examples; it does not fully identify conditional shift as the underlying cause. The real-data section mainly relies on recovery-based evidence. The argument is that if a small set of selected features can recover most of the performance loss, then the shift is sparse and interpretable. That is plausible, but it is also somewhat method-dependent.

Another issue is that the paper’s practical story is somewhat fragmented across variants. It's unclear when users should use SGShift, SGShift-A, or SGShift-K. The method family is coherent, but the practitioner needs more guidance, and the instruction could be sharper. The paper states that SGShift-K is for shifted-feature selection, while SGShift and SGShift-A can do simultaneous correction and selection, and that distinction should be more central in the presentation.

Some of the highest-level claims should be phrased a bit more cautiously. The strongest evidence is on controlled semi-synthetic data, and the real-data case studies are interesting demonstrations, but not definitive proof that the selected features uniquely explain true concept shift in the underlying scientific sense.

---

> ### Author Rebuttal · Authors · 2026-03-29
>
> Thank you for the accurate summary of our method and for highlighting SGShift’s innovation compared to prior work. The reviewer has raised 3 weaknesses and 3 questions. We summarize each review comment and provide a point-by-point response below.
>
> Please let us know if there are any additional questions or requests for further clarification. If the reviewer feels that their concerns have been adequately addressed, we would be grateful if they would consider updating their evaluation.
>
> > My main concern is about the real-world claims...
>
> jwpj-W1: Shift interpretation.
>
> Thank you for the comment. We agree that the proper interpretation is not that the underlying data generating mechanisms or causal processes are explicitly a sparse concept shift, but rather in the ML model sense these shifts can be closely represented by a sparse function. The proper interpretation is that there exists a high-fidelity explanation of the difference in distributions that depends on only a few variables. This interpretation is important and we will update the manuscript throughout in the camera ready version to make it clear this is how results should be viewed, and make it clear in the real data results we are presenting associative evidence and not causal.
>
> > Another issue is that the paper’s practical story is somewhat fragmented across variants..
>
> jwpj-W2: Variants
>
> Thank you for the suggestion. We agree that consolidating the story across variants is important for practitioners. Broadly, we recommend using SGShift-K for feature selection by default for most settings. SGShift-A may be more useful in feature selection with significant base model misfit, or as per Figure 2, low signal-to-noise regimes where it may outperform SGShift-K, but more generally is a good test for how much model performance can be recovered. Naive SGShift captures simpler shifts well enough, and serves as a strong baseline for future method development in diagnosing concept shift. We will add this practical guidance to the camera ready version.
>
> > Some of the highest-level claims should be phrased a bit more cautiously...
>
> jwpj-W3: Shift interpretation
>
> Please see our response to jwpj-W1. We will emphasize the real data interpretation is intended to be associative evidence.
>
> > Can the real-data claims be reframed more carefully?
>
> jwpj-Q1: Real data claims.
>
> Please see our response to jwpj-W3.
>
> > Can the paper provide a more explicit practical recommendation for when to use each of SGShift, SGShift-A, and SGShift-K?
>
> jwpj-Q2: Practical guidance
>
> Please see our response to jwpj-W2.
>
> >In the real-data experiments, is there a stronger baseline or alternative recovery strategy that could help show that the observed sparsity is not too method-specific?
>
> jwpj-Q3: Recovery strategies
>
> Thank you for the question. We recognize it is our responsibility to provide a clear exposition. While the baseline methods are able to do feature selection, only SGShift-A and SGShift are capable of doing this joint feature selection/performance recovery, as we directly learn terms to correct for the model. We believe SGShift alone is sufficient to show that these shifts can be modeled as a sparse concept shift, as per our updated interpretation of shifts in jwpj-W1. If any one method discovers a sparse correction term that recovers all performance, that indicates the shift can be represented sparsely, and another method failing to discover such a term would not deny the first method’s sparse representation. We will make this more clear in the corresponding section in the camera ready version.

---

> > ### Author Rebuttal · Reviewer_jwpj · 2026-04-02
> >
> > I thank the authors for their response. Some of my concerns have been partially addressed through the current rebuttal and the proposed revision plan. The authors clarify that, in real-world settings, the reported results should not be interpreted causally, but rather as guiding interpretation and shaping how such shifts are viewed. To be honest, I do not find this fully convincing in addressing the concern about real-world claims, particularly whether the observed shifts in real-world cases can truly be regarded as “true” shifts, but I will maintain my positive score for this work.

---

### Official Review · Reviewer_W6v5 · 2026-03-13

**Soundness:** 3
**Presentation:** 3
**Significance:** 3
**Originality:** 3
**Overall Recommendation:** 4
**Confidence:** 3

**Summary:**

This paper aims to explain concept shift (i.e., the reason why models trained in source domain failed for target domain) by pointing out those relationships between features and labels that have changed. The authors in turn of this problem to learning sparse, as the correction terms for the target domain given the source domain, SGShift, and find that the involved features in the correction process are the shifted features. Additional variants are proposed for source model misspecification (SGShift-A) and knockoff for false discovery control (SGShift-K). Experiments are performed on three semi-synthetic benchmarks on real medical datasets and two case studies on a real COVID-19 and a real Lupus case studies. SGShift demonstrates significant advantages over Diff, WhyShift, and SHAP in shifted feature identification, particularly SGShift-K.

**Compliance With Llm Reviewing Policy:**

Affirmed.

**Key Questions For Authors:**

See weaknesses.

**Limitations:**

yes

**Strengths And Weaknesses:**

Strengths

1.	The approach is also well formulated. The idea of casting concept shift explanation as sparse correction learning on source model is valid and a more natural interpretation compared to the dual-model subtraction, especially when target samples are sparse.
2.	The experiments are robust. In addition to the sparse shifts, the authors also test on denser shifts, small target samples and high dimension, strong correlations, low SNR. Overall the SGShifts family behaves well, and the SGShift-K particularly has the most powerful advantages.

Weaknesses

1.	Methodological novelty . We see novelty, but it's slight. Essentially we are ``putting existing pieces back together'' in terms of sparse regression, GAM and knockoff tools in the concept change explanatory setting. The methodological novelty is less clear.
2.	Much of the overall framework hinges on the premise that shifts are sparse . The authors also show good results under a dense shift, but this could indicate that the method is still robust as a feature rank ordering method but does not necessarily indicate how the true shift mechanism should have actually performed.
3.	It may further confirm the effectiveness of sparse correction instead of suggesting that real-world concept shifts are really sparse, as in almost all cases it turns out, it is sufficient to use a small sparse set of features to recover most of the performance loss in this real-world data. This interpretation seems somewhat premature.

---

> ### Author Rebuttal · Authors · 2026-03-29
>
> Thank you for the thoughtful summary of our work and for highlighting SGShift’s advantages over other methods. The reviewer has raised 3 weaknesses. We summarize each review comment and provide a point-by-point response below.
>
> Please let us know if there are any additional questions or requests for further clarification. If the reviewer feels that their concerns have been adequately addressed, we would be grateful if they would consider updating their evaluation.
>
> >Methodological novelty . We see novelty, but it's slight. Essentially we are ``putting existing pieces back together'' in terms of sparse regression, GAM and knockoff tools in the concept change explanatory setting. The methodological novelty is less clear.
>
> W6v5-W1: Novelty
>
> Thank you for the comment. We recognize it is our responsibility to provide a clear exposition. The novelty in our work lies less in using GAMs and knockoffs in a new setting, but rather framing concept shift as this sparse learnable process, and instantiate this learning with a few strong performing methods on tabular data with statistical guarantees. The key advantages of this framing over alternatives is that it does not require any causal priors or parametric knowledge of the dataset beforehand, SGShift can simply be drag-and-dropped to any tabular dataset. We will update the exposition in the introduction in the camera ready version to make this contribution more clear.
>
> >Much of the overall framework hinges on the premise that shifts are sparse . The authors also show good results under a dense shift, but this could indicate that the method is still robust as a feature rank ordering method but does not necessarily indicate how the true shift mechanism should have actually performed.
>
> W6v5-W2: Shift interpretation
>
> Thank you for the comment. We agree that the proper interpretation is not that the underlying data generating mechanisms or causal processes are explicitly a sparse concept shift, but rather in the ML model sense these shifts can be closely represented by a sparse function. The proper interpretation is that there exists a high-fidelity explanation of the difference in distributions that depends on only a few variables. This interpretation is important and we will update the manuscript throughout in the camera ready version to make it clear this is how results should be viewed, and make it clear in the real data results we are presenting associative evidence and not causal.
>
> >It may further confirm the effectiveness of sparse correction instead of suggesting that real-world concept shifts are really sparse, as in almost all cases it turns out, it is sufficient to use a small sparse set of features to recover most of the performance loss in this real-world data. This interpretation seems somewhat premature.
>
> W6v5-W3: Real data interpretation
>
> Please see our response to W6v5-W2 above.

---

> > ### Author Rebuttal · Reviewer_W6v5 · 2026-04-04
> >
> > I will maintain my positive score.

---

### Official Review · Reviewer_LgnL · 2026-03-13

**Soundness:** 3
**Presentation:** 3
**Significance:** 3
**Originality:** 3
**Overall Recommendation:** 4
**Confidence:** 3

**Summary:**

This paper proposes a method to identify a small set of features whose relationship with the label has changed and can explain the drop in the performance of a model from the source to the target domain. In particular, it proposes a framework to start with the source-trained model’s prediction as an offset, then add a sparse correction term to maximise the target-distribution performance. The authors provide three example uses of this framework, namely SGShift (lasso), SGShift-A (absorption to handle source misfit) and SGShift-K (knockoffs framework), along with theoretical guarantees. It is also accompanied by a range of experiments, including using synthetic and real-world datasets.

**Compliance With Llm Reviewing Policy:**

Affirmed.

**Key Questions For Authors:**

1. How do we choose the basis function $\phi(X)$? Is there possibly a data-driven procedure for selecting these functions automatically?
2. It is advised in the paper that for limited target-domain data, a simpler model for $\hat{\Delta}$ can be used.

(1) Is there a quantitative guideline on how simple this would be?

(2) What will happen to the performance if the model for $\hat{\Delta}$ is too simple or too complex?

3. In Theorem 3.2, the parameters $q$, $\alpha$ and $\tau$ are mentioned, but $\tau$ does not seem to be clearly defined in the main text. Could you define $\tau$ precisely, and if it is a hyperparameter, how is it set in practice?
4. How do we choose the hyperparameters, such as $\lambda_\omega$, $\lambda_\delta$ or $\lambda$, and how sensitive is the proposed method to these parameters? Are there default strategies that you would recommend to practitioners?
5. For clarity, can you state the dimensions of the variables and functions defined in the paper (e.g., X, phi, etc.) when they are introduced?

**Limitations:**

Yes

**Strengths And Weaknesses:**

Strengths:
1. The explanation of the problem formulation is clear.
2. The proposed method is a framework that can be used with any sparse regression methods, demonstrating its flexibility.
3. The authors provide theoretical guarantees of their method.
4. The experiment evaluation is quite extensive, with the proposed solution being competitive compared to the baselines (Diff, WhyShift and SHAP). Furthermore, the datasets include real-world settings, which nicely illustrate the interpretability benefits of the proposed method.

Weaknesses:
1. It is unclear how to choose the set of basis functions $\phi$. A default basis function is proposed, which is the identity function $\phi(X) = X$, but this restricts the model to a linear model, which might be too restrictive. While the interaction shift experiment provided in Appendix J suggests some robustness to underspecification, the paper does not provide practical guidance for selecting the basis function in real-world use.
2. It would be helpful to have clearer guidelines or default recommendations for hyperparameter selection (such as $\lambda_\omega$, $\lambda_\delta$ or $\lambda$) .

---

> ### Author Rebuttal · Authors · 2026-03-29
>
> Thank you for the accurate summary of our method and for highlighting SGShift’s flexibility and strong experimental results. The reviewer has raised 2 weaknesses and 5 questions. We summarize each review comment and provide a point-by-point response below.
>
> Please let us know if there are any additional questions or requests for further clarification. If the reviewer feels that their concerns have been adequately addressed, we would be grateful if they would consider updating their evaluation.
>
> > It is unclear how to choose the set of basis functions...
>
> LgnL-W1: Basis functions
>
> Thank you for the question. We recommend the default identity function unless the shift is expected to be more complex, in which case pairwise interaction terms can be added. Appendix J shows that adding many terms has little impact on performance, so it is likely unnecessary to choose specific interactions; including all pairwise terms should suffice. We acknowledge that kernel methods have been used to capture complex latent shift structure, but they may sacrifice the feature-level interpretability that SGShift provides. Developing basis functions that are both expressive and interpretable is an important direction for future work. We will add this guidance and future direction to the camera-ready version.
>
> > It would be helpful to have clearer guidelines or default recommendations for hyperparameter selection...
>
> LgnL-W2: Hyperparameter selection
>
> Thank you for the suggestion. We will add clearer default guidance for hyperparameter selection. For $\lambda$ (equivalently $\lambda_\delta$), our default recommendation is cross-validation on the target-domain loss; when sparse feature selection is the main goal, one may instead use the one-pass heuristic in Appendix E. For $\lambda_\omega$, we found performance to be relatively insensitive in practice, values in the range $0.1\lambda_\delta$ to $0.9\lambda_\delta$ worked well.
>
> > How do we choose the basis function ...
>
> LgnL-Q1: Basis functions
>
> Please see our response to LgnL-W1.
>
> > It is advised in the paper that for limited target-domain data, a simpler model for can be used.
> (1) Is there a quantitative guideline on how simple this would be?
> (2) What will happen to the performance if the model for
>  is too simple or too complex?
>
> LgnL-Q2: Simpler domain model
>
> Thank you for the questions. The reviewer raises two questions regarding (i) guidelines for shift modeling and (ii) performance if shift model is too simple/complex. We address each question in turn.
>
> (i) We instantiate SGShift with a relatively simple baseline to start with, L1 regularization with identity basis functions. We don’t anticipate that SGShift needs to be further simplified beyond that. We will make this clear in the camera ready version.
>
> (ii) If the shift model is too simple, SGShift may underdetect concept shift; if it is too complex, interpretability may suffer. Appendix J studies this in simulations where the basis functions are over- or under-specified under interaction shift. In both cases, SGShift still performs reasonably well despite the mismatch. Extending the method to domains such as images and text, which may require more complex shift models, is an important future direction.
>
> > In Theorem 3.2, the parameters... Could you define tau precisely, and if it is a hyperparameter, how is it set in practice?
>
> LgnL-Q3: Tau definition
>
> Tau is the knockoff threshold controlling the per-iteration false discoveries, as per Theorem 1 Equation 3.9 from the original knockoffs paper [1]. It is computed automatically from the knockoff statistics as recommended in that work (controlled by target $\alpha$ or $q$), and it need not be manually tuned. We will add this definition to the camera-ready.
>
> [1] Candes et al, Panning for Gold: ‘Model-X’ Knockoffs for High Dimensional Controlled Variable Selection, Journal of the Royal Statistical Society Series B: Statistical Methodology
>
> > How do we choose the hyperparameters ..., and how sensitive is the proposed method to these parameters? Are there default strategies that you would recommend to practitioners?
>
> LgnL-Q4: Hyperparameter selection.
>
> Thank you for the questions. The reviewer raises two questions regarding lambda’s (i) selection and (ii) stability, which we address in turn.
>
> (i) Please see our response to LgnL-W2.
>
> (ii) We performed additional analysis varying lambda to 0.1x to 10x the value selected in the main results. In all cases, the AUC did not differ by more than 0.03 from the results reported in Table 1, indicating SGShift is robust to this parameter selection. We will include this analysis in the camera-ready version.
>
> > For clarity, can you state the dimensions of the variables and functions defined in the paper (e.g., X, phi, etc.) when they are introduced?
>
> LgnL-Q5: Variable dimensions
>
> Thank you for the suggestion. We will make this update in the camera ready.

---

> > ### Author Rebuttal · Reviewer_LgnL · 2026-04-02
> >
> > Thank you for the clarification. My questions have been addressed, and my recommendation score remains positive.

---

### Decision · Program_Chairs · 2026-04-30

**Decision:**

Accept (regular)

**Comment:**

The reviewers are in agreement that the contributions proposed in this work are valuable to the community and should be published at ICML. They emphasize the clarity and novelty of the problem formulation, the generality of the method, the extensive experimental justification, and the theoretical guarantees as valuable and complimentary components of the work. The reviewers had originally expressed concerns about lack of information around the selection/justification of some of the components of the method (basis functions, hyperparameters), as they would impact the practical deployment of the method in the real world. The reviewers concerns were addressed sufficiently in the rebuttal. The ACs agree with the reviewers assessment and recommend acceptance, but encourage the authors to include the additional clarifications from the rebuttal in the camera ready to improve the narrative.